# Systems with Switching Causal Relations: A Meta-Causal Perspective

**Moritz Willig**[1,*]**, Tim Nelson Tobiasch**[1]**, Florian Peter Busch**[1,2]**, Jonas Seng**[1]**,
Devendra Singh Dhami**[3]**, Kristian Kersting**[1,2,4,5]

[1]Department of Computer Science, Technical University of Darmstadt, Germany
[2]Hessian Center for AI (hessian.AI), Germany
[3]Dept. of Mathematics and Computer Science, Eindhoven University of Technology, Netherlands
[4]Centre for Cognitive Science, Technical University of Darmstadt, Germany
[5]German Research Center for AI (DFKI), Germany
[*] moritz.willig@cs.tu-darmstadt.de

## Abstract

Most work on causality in machine learning assumes that causal relationships are driven by a constant underlying process. However, the flexibility of agents' actions or tipping points in the environmental process can change the qualitative dynamics of the system. As a result, new causal relationships may emerge, while existing ones change or disappear, resulting in an altered causal graph. To analyze these qualitative changes on the causal graph, we propose the concept of *meta-causal states*, which groups classical causal models into clusters based on equivalent qualitative behavior and consolidates specific mechanism parameterizations. We demonstrate how meta-causal states can be inferred from observed agent behavior, and discuss potential methods for disentangling these states from unlabeled data. Finally, we direct our analysis towards the application of a dynamical system, showing that meta-causal states can also emerge from inherent system dynamics, and thus constitute more than a context-dependent framework in which mechanisms emerge only as a result of external factors.

## 1 Introduction

Structural Causal Models (SCM) have become the de facto formalism in causality by representing causal relations as structural equation models (Pearl (2009); Spirtes et al. (2000)). While SCM are most useful for representing the intricate mechanistic details of systems, it can be challenging to derive the general qualitative behavior that emerges from the interplay of individual equations. Talking about the general *type* of relation emitted by particular mechanisms generalizes above the narrow computational view and has the ability to inspect causal systems from a more general perspective.

In addition to that, causal graphs can be subject to change whenever novel mechanisms emerge or vanish within a system. Prominently, agents can 'break' the natural unfolding of systems dynamics by forecasting system behavior and preemptively intervening in the course of events. As inherent parts of the environment, agents commonly establish or suppress the emergence of causal connections (Zhang & Bareinboim, 2017; Lee & Bareinboim, 2018; Dasgupta et al., 2019).

Consider the scenario shown in Figure 1 (left), where an agent $A$ (with position $A_X$) follows an agent $B$ (with position $B_X$) according to its internal policy $A_\pi$. We are interested in answering the question 'What is the cause of the current position of agent A?'. In general, the observed system can be formalized as follows: $A_X := f_A(B_X, U_A)$ and $B_X := f_B(U_B)$. Note that, from a classical causal perspective, we observe $B \rightarrow A$, since $A$ self-conditions itself to follow $B$, by instantiating the equation $f_A$ via its policy and thus becomes *dependent* on $B$. Classical causal considerations, which only consider how relations between variables are constructed, cannot give the correct answer. Only when we take a meta-causal stance and think about *how equation $f_A$ came to be* –how meta-causal states induce qualitative changes in behavior–, we can give a sufficient answer to this question.

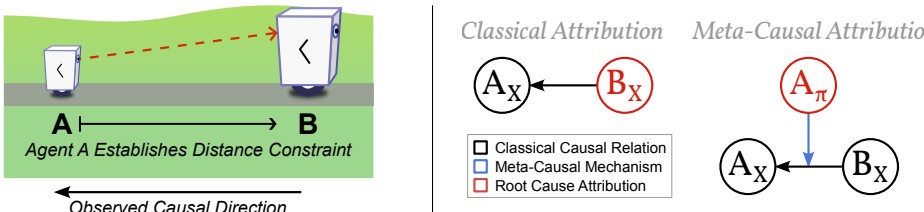

Figure 1: **Meta-Causality Identifies the Policy as a Meta Root Cause.** Agent $A$ intends to maintain its distance from agent $B$ by conditioning its position $A_X$ on the position $B_X$, which establishes a control mechanism, $A_X := f(B_X)$. In standard causal inference, we would infer $B_X \to A_X$ and, therefore, $B$ to be the root cause. Taking a meta-causal perspective reveals however, that $A_\pi$ establishes the edge $B_X \to A_X$ in the first place ($A_\pi \to (B_X \to A_X)$) such that $A_\pi$ is considered the root cause on the meta-level. *(Best Viewed in Color)*

Causal models do not exist in isolation, but emerge from the environmental dynamics of an underlying mediation process (Hofmann et al., 2020; Arnellos et al., 2006). Changes in causal relations due to intervention or environmental change are often assumed to be helpful conditions for identifying causal structures (Pearl, 2009; Peters et al., 2016; Zhang & Bareinboim, 2017; Dasgupta et al., 2019; Gerstenberg, 2024). These operations often work on the individual causal graphs, but never reason about the emerging meta-causal structure that governs the transitions between the different SCM. As a first formalization of metacausal models in this area, we describe how qualitative changes in the causal graphs can be summarized by metacausal models.

**Contributions and Structure.** The contributions of this work are as follows:

- To the best of our knowledge, we are the first to formally introduce typing mechanisms that generalize edges in causal graphs, and the first to provide a formalization of meta-causal models (MCM) that are capable of capturing the switching type dynamics of causal mechanisms.

- We present an approach to discover the number of meta-causal states in the bivariate case.

- We demonstrate that meta-causal models can be more powerful than classical SCM when it comes to expressing qualitative differences within certain system dynamics.

- We show how meta-causal analysis can lead to a different root cause attribution than classical causal inference, and is furthermore able to identify causes of mechanistic changes even when no actual change in effect can be observed.

We proceed as follows: In Section 2 we describe the necessary basics of Pearlian causality, mediation processes, and typing assumptions. In Sec. 3 we introduce meta-causality by first formalizing meta-causal frames, and the role of types as a generalization to a binary edge representations. Finally we define meta-causal models that are able to capture the qualitative dynamics in SCM behavior. In Sec. 4, we showcase several examples of meta-causal applications. Finally, we discuss connections to related work in Sec. 5 and conclude our findings in Sec. 6.

## 2   BACKGROUND

Providing a higher-level perspective on meta-causality touches on a number of existing works that leverage meta-causal ideas, even if not explicitly stated. We will highlight relations of these works in the 'Related Work' of section 5. Here, we continue to provide the necessary concepts on causality, mediation processes and typing, needed for the definitions in our paper:

**Causal Models.** A common formalization of causality is provided via Structural Causal Models (SCM; (Spirtes et al., 2000; Pearl, 2009)). An SCM is defined as a tuple $\mathcal{M} := (\mathbf{U}, \mathbf{V}, \mathbf{F}, P_\mathbf{U})$, where $\mathbf{U}$ is the set of exogenous variables, $\mathbf{V}$ is the set of endogenous variables, $\mathbf{F}$ is the set of structural equations determining endogenous variables, and $P_\mathbf{U}$ is the distribution over exogenous variables $\mathbf{U}$. Every endogenous variable $V_i \in \mathbf{V}$ is determined via a structural equation $v_i := f_i(\mathrm{pa}(v_i))$ that takes in a set of parent values $\mathrm{pa}(v_i)$, consisting of endogenous and exogenous

variables, and outputs the value of $v_i$. The set of all variables is denoted by $\mathbf{X} = \mathbf{U} \cup \mathbf{V}$ with values $\mathbf{x} \in \boldsymbol{\mathcal{X}}$ and $N = |\mathbf{X}|$. Every SCM $\mathcal{M}$ entails a graph $\mathcal{G}^{\mathcal{M}}$ that can be constructed by adding edges $(X_i, X_j) \in \mathbf{X} \times \mathbf{X}$ for each variable $X_j \in \mathbf{X}$ and its parents $X_i \in \text{Pa}(X_j)$. This can be expressed as an adjacency matrix $A \in \mathbb{B}^{N \times N}$ where $A_{ij} := 1$ if $(i, j) \in \mathcal{G}$ and $A_{ij} := 0$ otherwise.

**Mediation Processes.** Causal effects are embedded in an environment that governs the dynamics and mediates between different causes and effects. To reason about the existence of causal relations, we need to consider this process-embedding environment. We define a *mediation process* $\mathcal{E} = (\mathcal{S}, \sigma)$, adapted from Markov Decision Processes (Bellman (1957)) for our setting. Here, $\mathcal{S}$ is the state space of the environment and $\sigma : \mathcal{S} \to \mathcal{S}$ is the (possibly non-deterministic) transition function that takes the current state and outputs the next one. If we also have an initial state $s_0 \in \mathcal{S}$, we call $(\mathcal{E}, s_0)$ an *initialized mediation process*. As we are concerned with the general mediation process, we omit the common notion of a reward function $r$. Furthermore, we omit an explicit action space $\mathcal{A}$ and agents' policy $\boldsymbol{\pi}$ and model actions directly as part of the transition function $\sigma$. In accordance to considerations of Cohen (2022), this eases treatment of environment processes and agent actions as now both are defined on the same domain. The emergence of SCM from mediation processes can be studied under a measure-theoretic theory, as considered in Park et al. (2023). Similarly, Janzing & Mejia (2022) discuss the role of elementary actions towards the constitution of causal variables.

**Typing.** In the following section, we will make use of an identification function $\mathcal{I}$ to determine the presence or absence of edges between any two variables. In particular, one can make use of different identification functions to identify different *types* of edges. Previous work on typing causality exists (Brouillard et al., 2022), but primarily considers the types of variables that are causally related, rather than the type of structural relation itself. Other works in the field of cognitive science consider the perception of different types of mechanistic relations (e.g., 'causing', 'enabling', 'preventing', 'despite') based on the role that different objects play in physical scenes (Chockler & Halpern, 2004; Wolff, 2007; Sloman et al., 2009; Walsh & Sloman, 2011; Gerstenberg, 2022). Since all objects are usually governed by the same physical equations, this assignment of types serves to provide post hoc explanations of a scene, rather than to identify inherent properties of its computational aspects. Gerstenberg (2024) provides an 'intuitive psychology' example that fits well with our scenario.

## 3 META-CAUSALITY

Similar to Pearl & Mackenzie (2018), we define Meta-Causality to be *the [science of] change in qualitative cause-effect behavior.* Usually, the mediating process may be too fine-grained to yield interpretable models. Therefore, we consider a set of variables of interest $\mathbf{X}$ modeled by an SCM and introduced by an *abstraction* $\varphi : \mathcal{S} \to \boldsymbol{\mathcal{X}}$. (We provide a brief discussion on the emergence of SCM from mediating processes in Appendix A). Here, $\varphi$ could be defined as a summarization or causal abstraction over the state space (Rubenstein et al., 2017; Beckers & Halpern, 2019; Anand et al., 2022; Wahl et al., 2023; Kekić et al., 2023; Willig et al., 2023). In order to identify the type of causal relations from a mediating process, we need to be able to decide on what constitutes a type.

**Definition 1** (Meta-Causal Frame). *For a given mediation process $\mathcal{E} = (\mathcal{S}, \sigma)$ a **meta-causal frame** is a tuple $\mathcal{F} = (\mathcal{E}, \mathbf{X}, (\tau_{ij}), \mathcal{I})$ with:*

- *type-encoders $\tau_{ij} : \mathcal{X}_i \times \boldsymbol{\mathcal{X}}^{\mathcal{S}} \to \mathcal{T}$ that assign a **type** $t \in \mathcal{T}$ to the functional dependence of $X_j$ on $X_i$, induced by the underlying process $(\mathcal{S}, \sigma)$. This relation is between $\mathcal{X}_i$ (values of $X_i$) and the abstraction of the transition function $\varphi \circ \sigma \in \boldsymbol{\mathcal{X}}^{\mathcal{S}} = \{\psi : \mathcal{S} \to \boldsymbol{\mathcal{X}}\}$.*

- *an identification function $\mathcal{I} : \mathcal{S} \times \mathbf{X} \times \mathbf{X} \to \mathcal{T}$ with $\mathcal{I}(s, X_i, X_j) \mapsto t := \tau_{ij}(\varphi(s), \varphi \circ \sigma)$ that assigns a type to every pair of causal variables for any state of the environment.*

*Types* generalize the role of edges in causal graphs, while type encoders $\tau_{ij}$ determine the particular type of edges from properties of the underlying functional relations $\varphi \circ \sigma$. In most classical scenarios, the co-domain $\mathcal{T}$ of the type encoder $\tau_{ij}$ is chosen to be Boolean, representing the *existence* or *absence* of edges. In other cases, different values $t \in \mathcal{T}$ can be understood as particular types of edges, like positive, negative, or the absence of influence. This will help us to distinguish meta-causal states that share the same graph adjacency. The only requirement for $\mathcal{T}$ is that it must contain a special value *0*, which indicates the total absence of an edge. *Meta-causal states* now generalize the idea of binary adjacency matrices. We, intentionally, do not restrict the identification function in

any way. However, particular choices, e.g. functions identifying only direct causal effects, are more likely to result in classical SCM. We provide a short discussion on this in Appendix B.

**Definition 2** (Meta-Causal State). *In a meta-causal frame $\mathcal{F} = (\mathcal{E}, \mathbf{X}, \tau_{ij}, \mathcal{I})$, a **meta-causal state** is a matrix $T \in \mathcal{T}^{N \times N}$. For a given environment state $s \in \mathcal{S}$, the **actual meta-causal state** $T_s$ has the entries $T_{s,ij} := \mathcal{I}(s, X_i, X_j) = \tau_{ij}(\varphi(s), \varphi \circ \sigma)$.*

A meta-causal state $T \in \mathcal{T}^{N \times N}$ represents a graph containing edges $e_{ij}$ of a particular type $t \in \mathcal{T}$. In particular $T_{ij}$ indicates the presence ($T_{ij} \neq 0$) or absence ($T_{ij} = 0$) of individual edges. Our goal for meta-causal models (MCM) is to capture the dynamics of the underlying model. In particular, we are interested in modeling how different meta-causal states transition into each other. This behavior allows us to model them in a finite-state machine (Moore et al., 1956):

**Definition 3** (Meta-Causal Model). *For a meta-causal frame $\mathcal{F} = (\mathcal{E}, \mathbf{X}, \sigma, \mathcal{I})$, a **meta-causal model** is a finite-state machine defined as a tuple $\mathcal{A} = (\mathcal{T}^{N \times N}, \mathcal{S}, \delta)$, where the set of meta-causal states $\mathcal{T}^{N \times N}$ is the set of machine states, the set of environment states $\mathcal{S}$ is the input alphabet, and $\delta : \mathcal{T}^{N \times N} \times \mathcal{S} \to \mathcal{T}^{N \times N}$ is a transition function.*

Usually, we have the objective to learn the transition function $\delta$ for an unknown state transition function $\sigma$. The state transition function $\delta$ can be approximated as $\delta(T, s) := \mathcal{I}(\sigma'(T, s), X_i, X_j) = \tau_{ij}(\varphi(s), \varphi \circ \sigma'(T, s))$. In addition, $\sigma'$ takes the state $T$ to better approximate transition probabilities.

As standard causal relations emerge from the underlying mediation process, the meta-causal states emerge from different types of causal effects. The transition conditions of the finite-state machine are the configurations of the environment where the quality of some environmental dynamics represented by a type $t \in \mathcal{T}$ changes.

**Inferring the Meta-Causal States.** Even if the state transition function is known, it may be unclear from a single observation which exact meta-causal state led to the generation of a particular observation $S$. This is especially the case when two different meta-causal states can fit similar environmental dynamics. Even in the presence of latent factors (e.g., an agent's internal policy), the current dynamics of a system (e.g., induced by the agent's current policy) can sometimes be inferred from a series of observed environment states. This requires knowledge of the meta-causal dynamics, and is subject to the condition that sequences of observed states uniquely characterize the meta-causal state. Since MCM are defined as finite state machines, the exact condition for identifying the meta-causal state is that observed sequences are homing sequences (Moore et al., 1956). Note that the following example of a game of tag presented below exactly satisfies this condition, where the meta-causal states produce disjoint sets of environment states ('agent A faces agent B', or 'agent B faces agent A'), and thus can be inferred from either a single observation, when movement directions are observed, or two observations, when they need to be inferred from the change in position of two successive observations.

**'Game of Tag' Example.** Consider an idealized game of tag between two agents, with a simple causal graph and two different meta-states. In general, we expect agent $B$ to make arbitrary moves that increase its distance to $A$, while $A$ tries to catch up to $B$, or vice versa. In essence, this is a cyclic causal relationship between the agents, where both states have the same binary adjacency matrix. Note, however, that the two states differ in the *type* of relationship that goes from $A$ to $B$ (and $B$ to $A$). We can use an identification function that analyzes the current behavior of the agents to identify each edge. Since $A$ can tag $B$, the behavior of the system changes when the directions of the typed arrows are reversed, so that the type of the edges is either 'escaping' or 'chasing'.

While the underlying policy of an agent may not be apparent from observation as an endogenous variable, it can be inferred by observing the agents' actions over time. Knowing the rules of the game, one can assume that the encircling agent faces the other agent and thus moves towards it, while the fleeing agents show the opposite behavior:

$$(t_{B \to A} = \text{``A Chasing''}) \iff (\dot{A}_{pos} \cdot (B_{pos} - A_{pos}) > 0)$$

where $\dot{A}_{pos}$ is the velocity vector of agent $A$ (possibly computed from the position of two consecutive time steps); $A_{pos}$ and $B_{pos}$ are the agent positions, and $\cdot$ is the dot product. Once the edge types are known, the policy can be identified immediately.

**Causal and Anti-Causal Meta-Causal States.** Assigning meta-causal states to particular system observations can be understood as labeling the individual observations. However, it is generally

unclear whether the meta-causal state has an observational or a controlling character on the system under consideration. One could ask the question whether the system dynamics cause the meta-causal state, whether the meta-causal state causes the system dynamics, (or whether they are actually the same concept), similar to the well-known discussion "On Causal and Anticausal Learning" by Schölkopf et al. (2012), but from a meta-causal perspective.

At this point in time, we cannot give definitive conditions on how to answer this question, but we present two examples that support either one of two opposing views. First, in Sec. 4.3 we present a scenario of a dynamical system where the structural equations of both meta-causal states are the same, since the system dynamics are governed by its self-referential system dynamics. Intervening on the meta-causal state will therefore have no effect on the underlying structural equations, and thus can have no effect on the actual system dynamics. In such scenarios, the meta-causal is rather a descriptive label and cannot be considered an external conditioning factor. In the following, we discuss the opposite example, where a meta-causal state can be modeled as an external variable conditioning the structural equations.

**Role of Contextual Independencies.** Changes in the causal graph are often attributed to changes in the environment modeled by some exogenous variables $Z$, as, for example, leveraged with the invariant causal prediction (Peters et al., 2016). In this case, we get MCM where the transition function only contains self-loops. This makes it clear that MCM are a more general tool in analyzing meta-causal structures. If we introduce a 'no-edge' type $t = 0$ for this scenario, a meta-causal model can describe the condition that $X_i$ and $X_j$ are *contextually independent* for some $Z = z$. For a suitable type-encoder with a surjective mapping $\psi : \mathcal{Z} \to \mathcal{T}^{N \times N}$ and a given family of compatible (see Appendix C) decomposed structural equations $(f_{ij}^z)_{z \in \mathcal{Z}}$ we have $f_j^z := \bigcirc_{i \in 1..N} f_{ij}|Z$ and

$$f_{ij}|Z := \begin{cases} f_{ij}^z & \textbf{if } \psi(z)_{ij} \neq 0 \\ (< *) & \textbf{otherwise} \end{cases} \tag{1}$$

where $f_{ij}^z$ are the structural equations of the edge $e_{ij}$ that are active under the environment $Z$ and (in a slight imprecision in the actual definition) $(< *)$ is the function that carries on the previous function of the composition and discards $X_j$. While the 'no-edge' type $0$ could be handled like any other type, we have listed it for clarity such that individual variables $\mathbf{X}_i$ become *contextually independent* whenever $f_{ij}|Z = 0$. We provide conditions for the reduction of MCM in Appendix D.

## 4 APPLICATIONS

In this section, we discuss several applications of the meta-causal formalism. First, we revisit the motivational example and consider how meta-causal models can be used to attribute responsibility. Next, we identify the presence of multiple mechanisms for the bivariate case from sets of unlabeled data. Finally, we analyze the emergence of meta-causal states from a dynamical system, highlighting that meta-causal states are more expressive than simple conditionings of the adjacency matrix. Code is made available at `https://github.com/MoritzWillig/metaCausalModels`.

### 4.1 ATTRIBUTING RESPONSIBILITY

Consider again the motivational example of Figure 1, where an agent $A$ with position $A_X$ follows an agent $B$ with position $B_X$ as dictated by its policy $A_\pi$. In this scene, we can imagine a counterfactual scenario in which we replace the 'following' policy of agent $A$ with, e.g., a 'standing still' policy and find that the $B_X \to A_X$ edge vanishes. As a result, we infer the meta-causal mechanism $A_\pi \to (B_X \to A_X)$ for the system and thus $A_\pi$ as the root cause of for values of $A_X$. In conclusion, while $A$ is conditioned on $B$ on a low level, the meta-causal reason for the existence of the edge $B \to A$ is caused by the $A_\pi$. Both attributions, tracing back causes through the structural equations $A_X := f(B_X)$ or our meta-causal approach, are valid conclusions in their own regard.

Note that in this scenario, a classic counterfactual consideration, $A_X^{A_\pi=\text{standing still}} - A_X^{A_\pi=\text{following}}$, would also have inferred an effect of $A_\pi$ on $A_X$ from a purely value-based perspective. Attributing effects via observed changes in variable values is a valid approach, but it fails to explain preventative mechanisms. For example, consider the simple scenario where two locks prevent a door from opening. Classical counterfactual analysis would attribute zero effect to the opening of either lock.

| $k^*$ \ $k'$ | $d = 0.0$ | | | | | $d = 0.1$ | | | | | $d = 0.2$ | | | | |
|---|---|---|---|---|---|---|---|---|---|---|---|---|---|---|---|
| | - | 1 | 2 | 3 | 4 | - | 1 | 2 | 3 | 4 | - | 1 | 2 | 3 | 4 |
| 1 | 2 | 81 | 3 | 7 | 7 | 2 | 85 | 3 | 7 | 3 | 1 | 83 | 3 | 8 | 5 |
| 2 | 41 | 1 | 54 | 4 | 0 | 43 | 1 | 48 | 8 | 0 | 49 | 1 | 47 | 3 | 0 |
| 3 | 68 | 0 | 4 | 22 | 6 | 63 | 0 | 2 | 30 | 5 | 77 | 2 | 0 | 13 | 8 |
| 4 | 92 | 0 | 0 | 1 | 7 | 89 | 0 | 0 | 2 | 9 | 87 | 0 | 1 | 5 | 7 |

Table 1: **Confusion Matrices for Identifying Meta-Causal Mechanisms.** The table shows identification results for predicting the number of mechanisms for the bivariate case for 100 randomly sampled meta-causal mechanisms. $d$ is the maximum sample deviation from the average mechanism probability. Rows indicate the true number of mechanisms, while columns indicate the algorithms' predictions. '-' indicates setups where the algorithm did not make a decision. In general, the algorithm is rather conservative in its predictions. In all cases where a decision is made, the number of correct predictions along the diagonals dominate. The first and second most frequent predictions are marked in green and orange, respectively. *(Best Viewed in Color)*

A meta-causal perspective would reveal that the opening of the first lock already changes the meta-causal state of the model by removing its causal mechanisms that condition the state of the door. In a sense, our meta-causal causal perspective is similar to that of the actual causality framework (Halpern, 2016; Chockler & Halpern, 2004). However, actual causality operates at the 'actual', i.e. value-based, level and does not take the mechanistic meta-causal view into perspective. Meta-causal models already allow us to reason about causal effects after opening the first lock due to a change in the causal graph, without us having to consider the effects of opening the second lock at some point to reach that conclusion.

## 4.2 Discovering Meta-Causal States in the Bivariate Case

Our goal in this experiment is to recover the number of meta-causal states $K \in [1..4]$ from data that exists between two variables $X, Y$ that are directly connected by a linear equation with added noise. We assume that each meta-causal state gives rise to a different linear equation $f_k := \alpha_k X + \beta_k + \mathcal{N}, k \in \mathbb{N}$, where $\alpha_k, \beta_k$ are the slope and intercept of the respective mechanism and $\mathcal{N}$ is a zero-centered, symmetric, and quasiconvex noise distribution[1]. Without loss of generality, we apply Laplacian noise, for which an L1-regression can estimate the true parameters of the linear equations (Hoyer et al., 2008). The causal direction of the mechanism is randomly chosen between different meta-causal states. The exact sampling parameters and plots of the resulting distributions are described in Sec. G (and plots of sample distributions are shown in Fig. 3 in the Appendix). In general, this scenario corresponds to the setting described above of inferring the values of a latent variable $Z$ (with $K = |Z|$) that indicate a particular meta-causal state of the system. Our goal is to recover the number of parameterizations of the causal mechanisms, and as a consequence to be able to reason about which points were generated by which mechanism.

**Approach.** The problem we are trying to solve is twofold : first, we are initially unaware of the underlying meta-causal state $t$ that generated a particular data point $(x_i, y_i)$, which prevents us from estimating the parameterization $(\alpha_k, \beta_k)$ of the mechanism. Conversely, our lack of knowledge about the mechanism parameterizations $(\alpha_k, \beta_k)_{k \in [1..K]}$ prevents us from assigning class probabilities to the individual data points. Since neither the state assignment nor the mechanism parameterizations are initially known, we perform an Expectation-Maximization (EM; Dempster et al. (1977)) procedure to iteratively estimate and assign the observed data points to the discovered meta-causal state parameterizations. Due to the local convergence properties of the EM algorithm, we further embed it into a locally optimized local random sample consensus approach (LO-RANSAC; Fischler & Bolles (1981); Chum et al. (2003)). RANSAC approaches repeatedly sample initial parameter configurations to avoid local minima, and successively perform several steps of local optimization - here the EM algorithm - to regress the true parameters of the mechanism.

---

[1]Implying unimodality and monotonic decreasing from zero allows us to distinguish the noise mean and intercept and to recover the parameters from a simple linear regression

|  | Analysis Type | 1 Mechanism | 2 Mech. | 3 Mech. | 4 Mech. |
|---|---|---|---|---|---|
| Max. Class | Theoretical | 1 | 23 | 363 | 8,179 |
| Deviation = 0.0 | Empirical | 2 | 8 | 24 | 173 |
| Max. Class | Theoretical | 1 | 26 | 429 | 10,659 |
| Deviation = 0.1 | Empirical | 2 | 8 | 25 | 177 |
| Max. Class | Theoretical | 1 | 30 | 526 | 14,859 |
| Deviation = 0.2 | Empirical | 2 | 8 | 26 | 177 |

Table 2: **Estimated Number of Required Resamples for Obtaining a 95% Convergence Rate with the LO-RANSAC Algorithm per Number of Mechanisms and Maximum Class Deviation.** The empirical observed convergence rates of the EM algorithm drastically reduce the theoretical derived bound of required samples. This reveals that convergence assumptions where chosen to be quite conservative, and attests a good fit of the EM algorithm for regressing mechanism parameters.

Assuming for the moment that the correct number of mechanisms $k^*$ has been chosen, we assume that the EM algorithm is able to regress the parameters of the mechanisms, $\alpha_k^*, \beta_k^*$, whenever there exists a pair of points for each of the mechanisms, where both points of the pair are samples generated by that particular mechanism. The chances of sampling such an initial configuration decrease rapidly with an increasing number of mechanisms (e.g. $0.036\%$ probability for $k = 4$ and equal class probabilities). Furthermore, we assume that the sampling probabilities of the individual mechanisms in the data can deviate from the mean by up to a certain factor $d$. In our experiments, we consider setups with $d \in \{0.0, 0.1, 0.2\}$. Given the number of classes and the maximum sample deviation of the mechanisms, one can compute an upper bound on the number of resamples required to have a $95\%$ chance of drawing at least one valid initialization. The bound is maximized by assigning the first half of the classes the maximum deviation probability $P_{\text{k-max}} = (1 + d)/k$ and the other half the minimum deviation probability $P_{\text{k-min}} = (1 - d)/k$. We provide the formulas for the upper bound estimation and in Sec. E in the Appendix and provide the calculated required resample counts in Table 2. In the worst case, for a scenario with $k = 4$ mechanisms and $d = 0.2$ maximum class probability deviation, nearly $14,900$ restarts of the EM algorithm are required, drastically increasing the potential runtime.

In our experiments, we find that our assumptions about EM convergence are rather conservative. Our evaluations show that the EM algorithm is still likely to be able to regress the true parameters, given that some of the initial points are sampled from incorrect mechanisms. We measure the empirical convergence rate by measuring the convergence rate of the EM algorithm over 5,000 different setups (500 randomly generated setups with 10 parameter resamples each). We perform 5 EM steps for setups with $k = 1$ and $k = 2$ mechanisms, and increase to 10 EM iterations for 3 and 4 mechanisms. For each initialization, we count the EM algorithm as converged if the slope and intercept of the true and predicted values do not differ by more than an absolute value of $0.2$.

**Determining the Number of Mechanisms.** The above approach is able to regress the true parameterization of mechanisms when the real number of parameters $k^*$ is given. However, it is still unclear how to determine the correct $k^*$. Computing the parameters for all $k \in K$ and comparing for the best goodness of fit is generally a bad indicator for choosing the right $k$, since fitting more mechanisms usually captures additional noise and thus reduces the error. In our case, we take advantage of the fact that we assumed the noise to be Laplacian distributed. Thus, the residuals of the samples assigned to a particular mechanism can be tested against the Laplacian distribution.

The EM algorithms return the estimated parameters $\alpha_k', \beta_k'$, and the mean standard deviation $b_k'$ of the Laplacian[2]. This allows one to compute the class densities $k(x_i, y_i; \alpha_k', \beta_k', b_k')$ for a pair of values $(x_i, y_i)$. Since the assignment of mechanisms to a data point may be ambiguous due to the overlap in the estimated PDFs, we normalize the density values of all mechanisms per data point and consider only those points for which the probability of the dominant mechanism is $0.4$ higher than the second class: $(\mathbf{x}^j, \mathbf{y}^j) := ((x_i, y_i)|\#1 = j; f_i^{\#2} < P_i^{\#1} \times (1 - 0.4))$, where $\#n$ indicates the class with the n-th highest density value. Finally, the residuals $\mathbf{y}^j - f_k'^j(\mathbf{x}^j, \mathbf{y}^j; \alpha_k', \beta_k', b_k')$ are

---

[2]Since mechanisms can go in both directions, $X \to Y$ and $Y \to X$ (cyclic relations are not considered), we repeat the regression for both directions and use an Anderson-Darling test (Anderson & Darling, 1952) on the residual to test which of the distributions more closely resembles a Laplacian distribution at each step.

computed for all data points $(\mathbf{x}^j, \mathbf{x}^j)$ assigned to a particular predicted mechanism $f_k'^j$. We choose the parameterization that best fits the data for each $k \in [1..4]$ and, make use of the Anderson-Darling test (Anderson & Darling, 1952) to test the empirical distribution function of the residuals against the Laplacian distribution with an $\alpha = 0.95$ using significance values estimated by Chen (2002). If all residual distributions of all mechanisms for a given $k'$ pass the Anderson-Darling test, we choose that number as our predicted number of mechanisms. If the algorithm finds that none of the $k \in [1..4]$ setups pass the test, we refrain from making a decision. We provide the pseudo code for our method in Algorithm. 1 in the Appendix.

**Evaluation and Results.** We evaluate our approach over all $k \in [1..4]$ by generating 100 different datasets for every particular number of mechanisms. For every dataset we sample 500 data points from each mechanism $(x_i^k, y_i^k) = f_k(\alpha_k x_i^k + \beta_k + l_i)$ where $l_i \sim \mathcal{L}(0, b_k)$, using the same sampling method as before (c.f. Appendix G). Finally, the algorithm recovers the number of mechanisms.

We compare theoretically computed and empirical in convergence results in Table 2. In practice, we observe that the convergence of the EM algorithm is more favorable than estimated, reducing from 23 to 8 required examples for the simple case of $k = 2, d = 0.0$, and requiring up to 83-times fewer samples for the most challenging setup of $k = 4, d = 0.2$, reducing from a theoretical of 14,859 to an empirical estimate of 177 samples. The actual convergence probabilities and the formula for deriving to sample counts are given in Table 3 and Sec. F of the Appendix.

Table 1 shows the confusion matrices between the actual number of mechanism and the predicted number for different values of maximum class imbalances. In general, we find that our approach is rather conservative when in assigning a number of mechanisms. However, when only considering the cases where the number of mechanisms is assigned, the correct predictions along the main diagonal dominate with over 60% accuracy for $k = 4$ and $d \in \{0.0, 0.1\}$ and rising above 80% for $k \neq 4$ for. In the case of $d = 0.2$, the results indicate higher confusion rates with 41.6% accuracy for the overall worst case of $K = 4$.

**Extension to Meta-Causal State Discovery on Graphs.** Our results indicate that identifying meta-causal mechanisms even in the bivariate case comes with an increasing number of uncertainty when it comes to increasing numbers of mechanisms. Given that the number of mechanisms could be reliably inferred from data for all variables, the meta-causal states could be identified as all unique combinations of mechanisms that are jointly active at a certain point in time. To recover the full set of meta-causal states, one needs to be able to simultaneously estimate the triple of active parents for every mechanism, the mechanisms parameterizations, and the resulting meta-state assignment of all data points for every meta-state of the system. All of the three components can vary between each meta-causal state (e.g. edges vanishing or possibly switching direction, thus altering the parent sets). Note that whenever any of the three components is known, the problem becomes rather trivial. However, without making any additional assumptions on the model or data and given the results on the already challenging task of identifying mechanisms in the bivariate case, the extension to a unsupervised full-fledged meta-causal state discovery is not obvious to us at the time of writing and we leave it to future work to come up with a feasible algorithm.

## 4.3 A META-STATE ANALYSIS ON STRESS-INDUCED FATIGUE

Stress is a major cause of fatigue and can lead to other long-term health problems (Maisel et al., 2021; Franklin et al., 2012; Dimsdale, 2008; Bremner, 2006). While short-term exposure may be helpful in enhancing individual performance, long-term stress is detrimental and resilience may decline over time (Wang & Saudino, 2011; Maisel et al., 2021). While actual and perceived stress levels (Cohen et al., 1983; Bremner, 2006; Schlotz et al., 2011; Giannakakis et al., 2019) may vary between individuals (Calkins, 1994; Haggerty, 1996), the overall effect remains the same. We want to model such a system as an example. For simplicity, we present an idealized system that is radically reduced to the only factors of external stress, modeling everyday environmental factors, and the self-influencing level of internal/perceived stress.

While being rather simple in setup, the example serves as a good demonstration on how dynamical systems induce meta-causal states that exhibit qualitative different behavior, while employing the same set of underlying structural equations. As such, the inherent behavior of the system is not only due to a conditioning external factor. We use an identification function to distinguish between two different modes of operation of a causal mechanism. In particular, we are interested whether

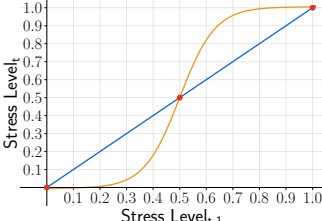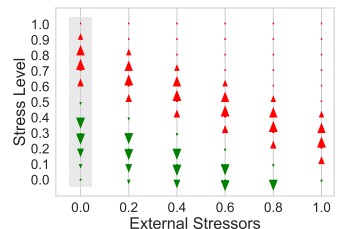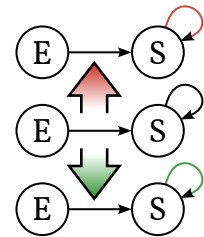

Figure 2: **Mechanistic Decomposition as Meta-Causal States**. **(left)** The effect of the stress level on itself (orange) plotted against the identity (blue; corresponding to a non-self-reinforcing effect). Once a certain threshold is reached, the function switches its behavior from self-suppressing to a self-reinforcing effect. **(center)** Contribution of the stress level mechanism for varying external stressors. Red arrows indicate a self-reinforcing effect, while green arrows indicate a suppressive effect. The gray area highlights the system configuration without external stressors. Although all states are governed by the same structural equations, our meta-causal analysis identifies the mechanistic difference and decomposes the corresponding initial conditions into different meta-causal states. **(right)** The standard SCM gets decomposed into different meta-causal states. While the graph adjacency remains the same, the different starting conditions identify different behavioral types of the causal mechanism. *(Best viewed in color.)*

the inherent dynamics of the internal stress level are self-reinforcing or self-suppressing. For easier analysis of the system we decompose the dynamics of the internal stress variable into a 'decayed stress' $d$ and 'resulting stress' $s$ computation. The first term are the previous stress levels decayed over time with external factors added. The resulting stress is then the output of a Sigmoidal function (Fig. 2, left) that either reinforces or suppresses the value (Fig. 2,center). The structural equations are defined as follows and we assume that all values lie in the interval of $[0, 1]$:

$$f_d := 0.95 \, \mathrm{clip}_{[0,1]}(s' + 0.5 \times \mathrm{ext.Stress}) \ \textbf{and} \ f_s := 1.01 \left( \frac{1}{1 + \exp(-15x + 7.5)} - 0.5 \right) + 0.5$$

where $s$ and $d$ are the resulting and decayed stress levels, and $s'$ is the previous stress level of $s$.

We now define the identification function to be $\mathcal{I} := \mathrm{sign}(\ddot{f}_s)$, where $\ddot{f}_s$ is the second order derivative of the Sigmoidal $f_s$, with either positive or negative effect on the stress level. The described system has two stable modes of dynamics. Note how the second-order inflection point at 0.5 of the Sigmoid acts as a transition point on the behavior of the mechanism. Stress values below 0.5 get suppressed, while values above 0.5 are amplified. Transitions between the two stable states can only be initiated via external stressors. Effectively this results in three possible meta-causal states which are governed via the following transition function:

$$\sigma : (t, s) \mapsto \begin{bmatrix} 1 & a \\ 0 & 0 \end{bmatrix} \text{ with } a := \mathrm{sign}(\ddot{f}_s) = \mathrm{sign}(s - 0.5) \in \{-1, 0, 1\}$$

**Role of Latent Conditioning.** A key takeaway of this example is that the current meta-state persists due to the inherent stress level and dynamics of the system. In contrast to other examples where system dynamics where purely due to the meta-causal state, variable *values* play a role in the overall system dynamics. As a result, the stressed state of a person would persist even when the initiating external stressors disappear. Intervening on the meta-causal state of the system is now ill-defined, as both positive and negative reinforcing effects are governed by the same equation. Thus, creating a disparity between the intervened meta-causal state and the systems' identified functional behavior.

## 5   RELATED WORK

**Causal Transportability and Reinforcement Learning.** Meta-causal models cover cases that reduce to conditionally dependent causal graphs due to changing environments (Peters et al., 2016; Heinze-Deml et al., 2018), but also extend beyond that for dynamical systems. In this sense, the work of Talon et al. (2024) takes a meta-causal view by transporting edges of different causal effects between environments. In general, the transportability of causal relations (Zhang & Bareinboim,

2017; Lee & Bareinboim, 2018; Correa et al., 2022) can be thought of as learning identification functions that identify general conditions of the underlying processes to transfer certain types of causal effects between environments Sonar et al. (2021); Yu et al. (2024); Brouillard et al. (2022). This has been studied to some extent under the name of meta-reinforcement learning, which attempts to predict causal relations from the observations of environments Sauter et al. (2023); Dasgupta et al. (2019). Generally, transferability has been considered in reinforcement learning, where the efficient use of data is omnipresent and causality provides guarantees regarding the transferability of mechanisms between environments (Sæmundsson et al., 2018; Dasgupta et al., 2019; Zeng et al., 2023).

**Gating Models.** Modeling switching causal relations (Liu et al., 2023) via so called 'gates' has been considered in prior works Minka & Winn (2008); Winn (2012). While MCMs extend beyond context-specific independencies, gates pose a practical way of modeling switching relations in cases where MCMs can be reduced to context-conditioned SCM.

**Large Language Models (LLMs).** Meta-causal representations are an important consideration for LLMs and other foundation models, since these models are typically limited to learning world dynamics from purely observational textual descriptions. LLMs need to learn meta-causal models that allow them to simulate the consequences of interventions (Lampinen et al., 2024; Li et al., 2021; 2022). To the best of our knowledge, Zečević et al. (2023) made the first attempt to define explicit meta-causal models that integrate with the Pearlian causal framework. However, their MCM are purely defined as adjacency matrix memorization, such that causal reasoning in LLMs equals a simple knowledge recall $(X \rightarrow Y) \Leftrightarrow (e_{XY} \in \text{Mem})$ and thus fails to generalize to novel scenarios.

**Actual Causality, Attribution and Cognition.** Our MCM framework can be used to infer and attribute responsibility, as shown in Sec. 4.1, and therefore touches on the topics of fairness, *actual causation* (AC) and work on counterfactual reasoning in cognitive science. (Von Kügelgen et al., 2022; Karimi et al., 2021; Halpern, 2000; 2016; Chockler & Halpern, 2004; Gerstenberg et al., 2014; Gerstenberg, 2024). The similarities also extend to how MCM encourage reasoning about the dependence between actual environmental contingencies and qualitative types of causal mechanisms. In this sense, MCM allow for the direct characterization of actual causes of system dynamics types, but a rigorous formalization is open for future work. While AC in combination with classical SCM only describes relationships between causal variable configurations and observable events, there are cases where we can take an MCM and derive an SCM that encodes the meta-causal types with instrumental variables, thus allowing a similar meta-causal analysis within the AC framework.

## 6    CONCLUSION

We formally introduced meta-causal models that are able to capture the dynamics of switching causal types of causal graphs and, in many cases, better express the qualitative behavior of the system under consideration. Within MCM, types generalize the notion of specific structural equations and abstract away unnecessary detail. We presented a motivating example of how a classical causal and a meta-causal inference might disagree on the attribute of root causes. We extended claims by considering that MCM are still able attribute changes in mechanistic behavior of a system, even when no actual changes becomes apparent. We presented a first approach to recover meta-causal states in the bivariate case. Although our experimental results only represent a first preliminary approach, we find that MCM are a powerful tool for modeling, reasoning, and inferring system dynamics. We demonstrated how MCM can be deployed dynamical systems and proved that they extend beyond conventional SCM.

**Limitations and Future Work.** While we have formally introduced meta-causal models, there remain several open directions to pursue, which we briefly touch on in Appendix H. We have been able to provide examples that illustrate the differences between standard causal, and meta-causal attribution. In particular, the combined application of standard and meta-causal explainability will allow for the joint consideration of actual and mechanistic in future attribution methods. However, our approach to recovering meta-causal states from unlabeled data is open to extension. Discovery on the full causal graphs is a desirable goal that is difficult to achieve for the reasons discussed in this paper. Finally, we made a first attempt to present examples for and against the controlling or observational role of meta-causal models, which we briefly discuss further in Appendix I. While, the presented observational perspective on MCM is mainly of interest for analytical applications, the application to agent systems and reinforcement learning might open up further fields of application.

ACKNOWLEDGMENTS

The Technical University of Darmstadt authors received funding by the EU project EXPLAIN, funded by the German Federal Ministry of Education and Research (BMBF) (grant 01—S22030D). They acknowledge the support of the German Science Foundation (DFG) project "Causality, Argumentation, and Machine Learning" (CAML2, KE 1686/3-2) of the SPP 1999 "Robust Argumentation Machines" (RATIO). This work is supported by the Hessian Ministry of Higher Education, Research, Science and the Arts (HMWK; projects "The Third Wave of AI"). It was funded by the European Union. Views and opinions expressed are, however, those of the author(s) only and do not necessarily reflect those of the European Union or the European Health and Digital Executive Agency (HaDEA). Neither the European Union nor the granting authority can be held responsible for them. Grant Agreement no. 101120763 - TANGO. This work was supported from the National High-Performance Computing project for Computational Engineering Sciences (NHR4CES).

The Eindhoven University of Technology authors received support from their Department of Mathematics and Computer Science and the Eindhoven Artificial Intelligence Systems Institute.

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

APPENDIX FOR "SYSTEMS WITH SWITCHING CAUSAL RELATIONS: A META-CAUSAL PERSPECTIVE"

The appendix is structured as follows: In Sec. A we describe the general emergence of causal relations in SCM from the underlying mediating process. In Sec. B we discuss desirable classes of identification functions. In Sec. C we consider the existence and compatibility of decomposed equations. In Sec. D we prove a condition for the reduction for meta-causal models to conditioned SCM. In Sec. E we derive the formula for the theoretical upper bound of the $95\%$ confidence interval. Sec F present the sample statistics and convergence results of the applied LO-RANSAC algorithm. In Sec. G we present the exact parameters for sampling bivariate meta-causal mechanism data. In Sec. H we discuss further practical applications of meta-causal models. Finally, in Sec. I, we discuss possible assertive properties of meta-causal models.

## A    EMERGENCE OF CAUSAL EFFECTS FROM MEDIATING PROCESSES

The definition of a meta-causal frames (Def. 1) grounds the emergence of standard SCM to an underlying mediating process $\mathcal{E}$. In particular, for any meta-causal frame, particular causal interactions are read via the identification function. While the underlying mediation process is time-dependent, the resulting causal graph is the projection of all causal interactions within the process onto a graphical structure. Note, however, that the resulting SCM still preserves the sequence of causal interactions through the DAG induced partial ordering of variables. This 'logical' time ordering (induced via the partial order) can be seen as an abstraction of the underlying process.

## B    DESIRABLE CLASSES OF IDENTIFICATION FUNCTIONS

Particular choices of different identification functions will result in different identified meta-causal states. However, from a classical causal perspective it might be desirable to choose particular classes of functions to identify faithful SCM. In particular, one could ask the question whether indirect effects are of interest and should also be identified by the identification function. Consider a scenario where three variables $X, Y, Z$ are considered as part of an to-be-identified SCM from an underlying process. In our scenario we identify the direct causal effects $X \rightarrow Y$ and $Y \rightarrow Z$ and –under the assumption that no further direct edges can be identified–, assume the graph to be faithful and we do not identify the indirect causal relation $X \rightarrow Z$. This however changes, once variable $Y$ is dropped out of the variable set $\mathbf{X}$ of the SCM. Now, with the same underlying mediating process we want our identification function to identify $X \rightarrow Z$, which was omitted before. Generally, we assume that in most cases identification function that identify direct causal effects with regard to $\mathbf{X}$ are the most common.

Following this rather high-level discussion, we frame the previous procedure in terms of the identification function and functional relation. Given all functional relations $\mathcal{X}^S$, the type decoder can determine whether the whole functional relation $\mathcal{X}_j$ between variables $X_j$ is mediated by some other (set of) intermediate mechanism(s) $\mathcal{X}_{\mathbf{z}}$[3]. In situations where $X_i \rightarrow X_{\mathbf{z}} \rightarrow X_j$ the type encoder needs to identify whether the relation $X_i \rightarrow X_j$ is purely mediated via $\mathcal{X}_{\mathbf{z}}^S$ (meaning that all computational paths from $X_i$ to $X_j$ in $\mathcal{X}_j^S$ that does not passes through $\mathcal{X}_{\mathbf{z}}^S$) or some additional direct effect exists that are not 'shielded' by $\mathcal{X}_{\mathbf{z}}^S$. Conversely, whenever some $X_k \in X_{\mathbf{z}}$ of the intermediate set is dropped/marginalized from the causal variables, the mechanism $\mathcal{X}_k^S \subset \mathcal{X}_j^S$ is no longer identified as the mechanism of a causal variable within the overall $\mathcal{X}_j^S$ and (given that the remaining $\mathcal{X}_{\mathbf{z}\setminus\{k\}}^S$ does not shield $X_i$ from $X_j$) a direct arrow $X_i \rightarrow X_j$ can be inferred.

Given our meta-causal framework we, however, have the freedom to identify the indirect relations. In that case one might assign these edges a particular '[1-hop] indirection' type, indicating it to

---

[3]Note in this context, that we assume structural equations to be uniquely identifiable. This might be implemented via type-theoretic considerations in which every variable of the underlying process gets assigned a unique type, such that for every $s_i \in \mathcal{S}_i$, $\forall i, j.i \neq j \Rightarrow \mathcal{S}_i \neq \mathcal{S}_j$ hold. Even though two functions might be isomorphic they can be thus be differentiated via their domain and codomain. In practice, one might additionally consider the computations graph of the mediation process to uniquely differentiate between isomorphic functions

be the result of an indirect relation via an (possibly unobserved) intermediate variable. Consider, how this notion of edge types over unobserved variables is already in use in CPDAGs –as for example produced by the PC or GES algorithms (Spirtes et al., 2000; Chickering, 2002)– to indicate undirected edges for which the orientation is currently unknown.

## C    COMPATIBILITY OF DECOMPOSED EQUATIONS

In this section, we provide a brief discussion on the existence and compatibility of functions $X$ as considered in Equation 1.

Since the conditioned SCM in Eq. 1 is modeled after an the particular causal model that exists under a meta-causal state indexed by $z$, it follows that a particular composition of functions $f_{ij}^z$ has to exist (and that the functions are compatible), since the full function $f_j^z$ exists for each meta-causal state. In a naïve approach, the order of composition $i \in \{1..N\}$ enforces a particular evaluation order of the functions, and in particular requires this order to be the same for every meta-causal state.

Generally, the presented function composition might not work, in the case that the individual functions get chosen badly from the start. A simple solution to overcome such problem is to assume compositions of lifted functions and assume their signatures to be compatible (which is always permitted due to the known existence of the composed $f_j^z$). Note that functions $f_{ij}^z, f_{ij}^{z'}$, whose functional type $t_{ij}$ did not change under a change of $z$ to $z'$, to make signatures compatible.

**Common handling of compositionality in SCM:** The adjustment of signatures is in fact often considered in the case of compositional SCM, e.g. additive noise models, where the signature of an outer 'merging function' $f_j(f_{1j}, \ldots, f_{Nj})$, e.g. $\sum_{i \in pa(j)} f_{ij}$, is in fact adjusted based on the number of parents (or, otherwise, zero weight edges are incorrectly excluded from the parent set).

## D    REDUCTION OF META-CAUSAL MODELS TO CONDITIONED SCM

In this section, we provide a condition under which meta-causal models can be reduced to ordinary SCM with structural equations conditioned on some external variable as related to Eq. 1.

**Definition 4** (MCM Reducability). *For a given mediation process $\mathcal{E} = (\mathcal{S}, \sigma)$ and abstraction $\varphi : \mathcal{S} \to \boldsymbol{\mathcal{X}}$ we call a meta-causal model $\mathcal{A} = (\mathcal{T}^{N \times N}, \mathcal{S}, \delta)$ **reducible** to a conditioned SCM if there is some SCM over $\mathbf{X} \cup \{Z\}$ where the types of functional dependencies between $X_i, X_j \in \mathbf{X}$ can be fully determined by $Z$.*

**Theorem D.1** (Specific Criterion for Meta-Causal Reducability). *If for a given mediation process $\mathcal{E} = (\mathcal{S}, \sigma)$ and abstraction $\varphi : \mathcal{S} \to \boldsymbol{\mathcal{X}}$, a meta-causal model $\mathcal{A} = (\mathcal{T}^{N \times N}, \mathcal{S}, \delta)$ all its transitions are loops, then it is reducible to a conditioned SCM.*

*Proof.* If a meta-causal model only has loops as transitions, the meta-causal types are independent of the modeled mediation process. While types can differ for different starting conditions in the environment, its transition process never results in a type change. Hence, meta-causal types induce an equivalence relation on $\mathcal{S}$, compatible with $\sigma$ and we can introduce an exogenous variable $Z$ with $\mathcal{Z} = \mathcal{T}^{N \times N}$ or, alternatively, with values $\mathcal{Z}$ for each connected component in $\mathcal{E} = (\mathcal{S}, \sigma)$. $\qquad \square$

We also see some potential to weaken this criterion and, therefore, find a more general condition for reducibility by further examination of the abstraction that links the mediating process with the causal variables for future work. The primary obstacle in this regard is that meta-causal models abstract away some information about structural equations, such that interventions on $Z$ might lead to a mismatch between the resulting SCM and the underlying mediation process.

## E    PROBABILITIES FOR SAMPLE COMPUTATION AND UPPER BOUNDS

Consider a dataset $\mathcal{D} \in \mathbf{R}^{m \times 2}$ of $m$ samples over two variables where we want to separate $n$ different functions. We assume that the data distribution contains a uniform number of samples from each function, where each class could be under- or overrepresented by an offset of $d$. Specifically, we

assume that each function is represented by $(1\pm\mathrm{d})\frac{1}{n}|\mathcal{D}|$ samples, i.e., the probability of encountering a particular class $X$ is $\frac{1-\mathrm{d}}{n} \leq P(X) \leq \frac{1+\mathrm{d}}{n}$ and $\mathbb{E}[P(X)] = \frac{1}{n}$. To identify all functions between these two variables, we assume linearity and apply EM with RANSAC on random pairs of samples (see Section 4.2). By selecting $n$ pairs, there is a chance that one pair is chosen from each function (we will refer to such a set of pairs as a "correct" set of samples). In this section, we derive the probability of a correct pair being chosen at random, so that we can estimate how many times pairs need to be sampled to reliably encounter a correct set of samples.

We denote $S$ as the event of "correctly" sampling all $n$ pairs from all $n$ different functions. If all classes have the same number of samples, the chance of randomly selecting a pair from a new class is $\frac{n}{n} \cdot \frac{1}{n}$ for the first pair of samples, $\frac{n-1}{n} \cdot \frac{1}{n}$ for the second, ..., and $\frac{1}{n} \cdot \frac{1}{n}$ for the last; in short:

$$\mathbb{E}[P_n(S)] = \frac{n!}{n^n} \cdot \left(\frac{1}{n}\right)^n = \frac{n!}{n^{2n}}. \tag{2}$$

If the data distribution is not perfectly uniform, i.e., $\mathrm{d} \neq 0$, we can also calculate a lower bound for the same probability. Consider two probabilities per sample: the **probability of selecting a new class** $P_n(S^{\mathrm{new}})$ and the **probability of selecting a second sample of the same class** $P_n(S^{\mathrm{same}})$ afterwards. Across all samples, these correspond to $\frac{n!}{n^n}$ and $(\frac{1}{n})^n$ in $\mathbb{E}[P_n(S)]$, respectively.

Let us first consider the **probability of selecting a new class**. When the first sample is taken, only one new class can be selected (probability of 1). If this sample was taken from the largest class first, the probability that subsequent samples will be taken from new classes decreases, since the space of "unsampled" classes is smaller. For this lower bound, we therefore assume that maximally large classes are sampled from as much and as early as possible. According to our assumptions, the largest classes each take up a fraction of $\frac{1+\mathrm{d}}{n}$ of the data. Therefore, the probability of selecting a new class for successive samples has the following probabilities $\frac{n}{n}, \frac{n-(1+\mathrm{d})}{n}, \frac{n-(1+\mathrm{d})2}{n}, \ldots$. First, consider the case where $n$ is even. Here, after all the $\frac{n}{2}$ largest classes have been selected, only the small classes remain. For the last, second to last, ... classes, this probability is represented by $\frac{(1-\mathrm{d})}{n}, \frac{2(1-\mathrm{d})}{n}, \ldots$. Overall, we get the probability

$$P_n(S_{\mathrm{even}}^{\mathrm{new}}) \geq \left(\prod_{i=\frac{n}{2}}^{n} \frac{n - (1+\mathrm{d})(n-i)}{n}\right) \left(\prod_{i=1}^{\frac{n}{2}-1} \frac{i(1-\mathrm{d})}{n}\right).$$

If $n$ is uneven, an average size class between the largest and smallest classes must be included:

$$P_n(S_{\mathrm{odd}}^{\mathrm{new}}) \geq \left(\prod_{i=\frac{n}{2}+0.5}^{n} \frac{n - (1+\mathrm{d})(n-i)}{n}\right) \left(\frac{n - (1+\mathrm{d})(\frac{n}{2}-0.5) - 1}{n}\right) \left(\prod_{i=1}^{\frac{n}{2}-1.5} \frac{i(1-\mathrm{d})}{n}\right).$$

The **probability of selecting a second sample of the same class** is easier to calculate. Instead of constant probabilities as in the expectation with $\frac{1}{n}$, we now have two different probabilities in the even case and three in the odd case. In the even case, we have $\frac{n}{2}$ large batches and the same number of small batches, so the probability of choosing the right batch each time is

$$P_n(S_{\mathrm{even}}^{\mathrm{same}}) \geq \left(\frac{1+\mathrm{d}}{n}\right)^{\frac{n}{2}} \left(\frac{1-\mathrm{d}}{n}\right)^{\frac{n}{2}}.$$

In the uneven case, we also have to also consider the batch that has an average size

$$P_n(S_{\mathrm{odd}}^{\mathrm{same}}) \geq \left(\frac{1+\mathrm{d}}{n}\right)^{\frac{n}{2}} \frac{1}{n} \left(\frac{1-\mathrm{d}}{n}\right)^{\frac{n}{2}}.$$

Note that for both $P_n(S_{\mathrm{new}})$ and $P_n(S_{\mathrm{same}})$, the distribution of the data into the largest and smallest possible batches (according to our assumptions) results in the smallest possible probabilities; hence, the computed probability is a lower bound. If the batches were more evenly sized, the probability would be larger. We can also see that a deviation of up to 1 results in a probability of 0, since it is impossible to sample from a class that is not represented in the data.

| Class Deviation | 1 Mechanism | 2 Mechanisms | 3 Mechanisms | 4 Mechanisms |
|---|---|---|---|---|
| 0.0 | 4219 (84.38%) | 1740 (34.80%) | 592 (11.84%) | 86 (1.72%) |
| 0.1 | " | 1702 (34.04%) | 577 (11.54%) | 84 (1.68%) |
| 0.2 | " | 1567 (31.34%) | 555 (11.10%) | 84 (1.68%) |

Table 3: **Empirical estimated convergence percentage of the EM algorithm for different class imbalances and number of mechanisms.** The table shows the number of samples converged and the convergence rates (in parentheses) for a single random initialization and for estimating the parameterization of the underlying system for a given number of mechanisms. All results are reported per 5000 samples.

In total, the probability of selecting of a correct sample set is the product of the two probabilities above, i.e,

$$P_n(S) = \begin{cases} P_n(S_{\text{even}}^{\text{new}})P_n(S_{\text{even}}^{\text{same}}) & \text{if } n \text{ is even} \\ P_n(S_{\text{odd}}^{\text{new}})P_n(S_{\text{odd}}^{\text{same}}) & \text{if } n \text{ is odd.} \end{cases} \quad (3)$$

$P_n(S)$ is the (lower bound on the) probability of selecting a correct set of samples. We can calculate the number of trials $k$ needed to find such a set of samples with at least 95% probability. The opposite probability, of never finding it with less than 5% probability, is easier to calculate:

$$(1 - P(S))^k \leq 1 - 0.95 = 0.05$$
$$k \ln(1 - P(S)) \leq \ln(0.05)$$
$$k \leq \frac{\ln(0.05)}{\ln(1 - P(S))}$$

This allows us to determine how many attempts might be necessary. Note that while this would leave a 5% chance of not picking the right samples, there are various practical reasons why the actual probability of finding a working set of samples will be higher, e.g., if the number of samples from each class is not as uneven as assumed, or if some samples are distributed in such a way that even picking a sample from the "wrong" class might still lead to the identification of the correct mechanisms.

For example, if $n = 2$ and $\mathrm{d} = 0.2$, we have an expected probability of $\mathbb{E}[P(S)] = \frac{2!}{2^{2 \cdot 2}} = 0.125$ and a lower bound of $P_2(S) = P(S_{\text{even}}^{\text{new}})P(S_{\text{even}}^{\text{same}}) \geq \frac{0.8}{2} \cdot \frac{2}{2} \cdot \frac{1.2}{2} \cdot \frac{0.8}{2} = 0.096$. Larger deviations decrease the probability while a deviation of $\mathrm{d} = 0$ results in the same probability as with $\mathbb{E}[P(S)]$. For the lower bound, this results in $k = 30$ samples for a confidence of 95% using the above calculation steps. All resulting sample counts can be found in Table 2.

## F   EM CONVERGENCE RESULTS

The required number of resamples for a 95% success rate of the RANSAC algorithm is calculated by $\log(0.05)/\log(1 - C^1)$, where $S1$ are the convergence rates for the individual samples computed in Sec. E.

Empirical convergence probabilities and resulting resampling counts for the LO-RANSAC algorithm are shown in Tables 2 and 3. Table 4 lists the goodness of fit for all converged samples. In general, we find that in cases where the approach is able to converge, it undercuts the required parameter convergence boundary of 0.2 by factors of 4.8 and 3.5 for the slope and intercept, respectively.

## G   MECHANISM SAMPLING

For our experiments in Sec. 4.2 we uniformly sample the number of mechanisms to be in $K \in \{1..4\}$. The slopes of the linear equations are uniformly sampled between $\alpha \in \pm[0.2..5]$ and the intercepts are in the range $\beta \in [-5, 5]$. We add Laplacian noise $\mathcal{L}(x|\mu, b) = \frac{1}{2b} * \exp(-\frac{|x-\mu|}{b})$ with $\mu = 0$ and $b \in [0.1, 4.0]$. $X$ values are uniformly sampled in the range $[-5, 5]$ and $y_i = \alpha x_i + \beta + \mathcal{L}(x|0, b)$. The average number of samples per class is set to 500. Throughout the experiments,

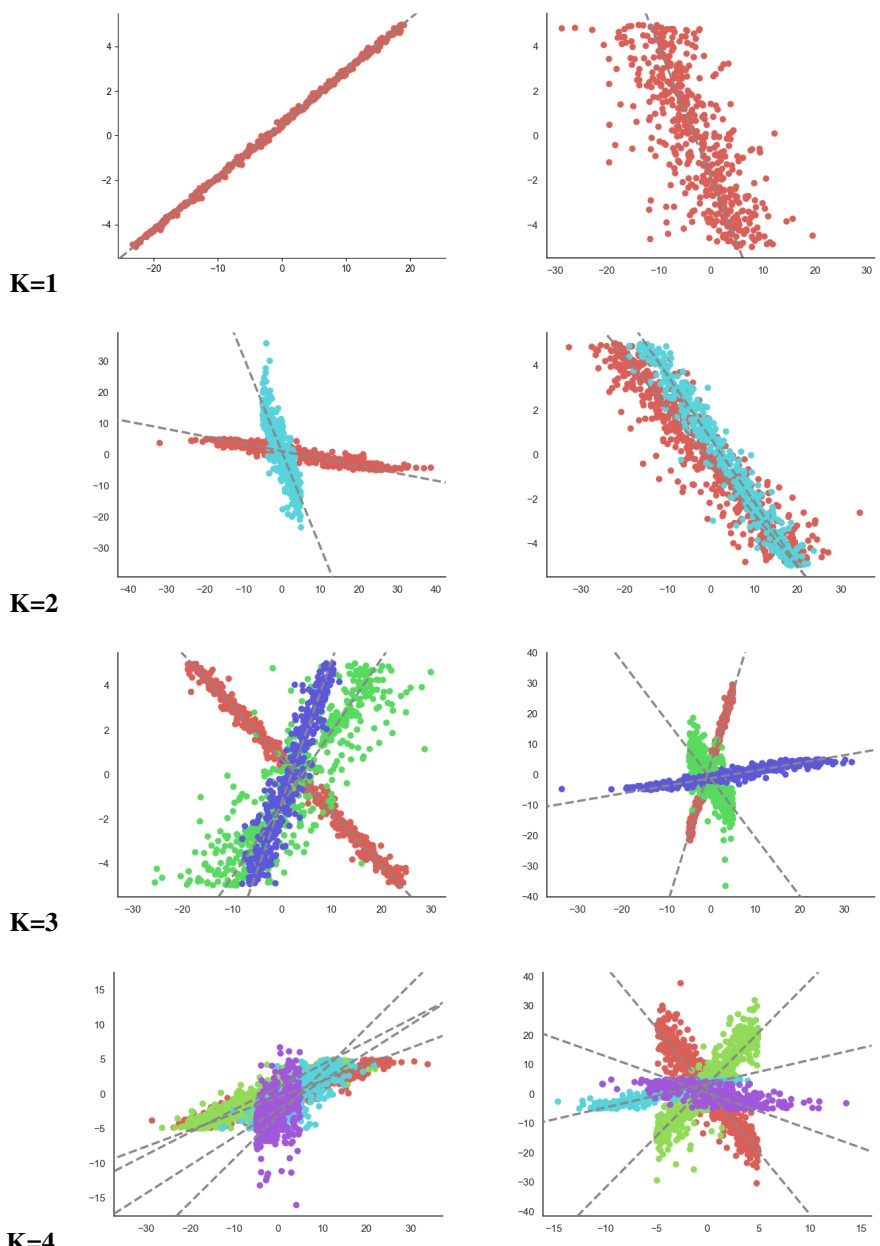

Figure 3: **Sampled Mechanisms.** The figure shows a selection of different randomly sampled mechanism distributions, ranging from one to up to four simultaneously present mechanisms. The gray dotted lines represent the generating ground truth mechanisms. *(Best Viewed In Color)*

we vary the class probabilities by a class deviation factor $D \in \{0.0, 0.1, 0.2\}$. Specifically, we maximize the class deviation by assigning $K/2$ classes the maximum probability $1/K * (1 + D)$ and $K/2$ classes the minimum probability $1/K * (1 - D)$. If $K$ is odd, a class is assigned the average probability $1/K$. We show a selection of the resulting sample distributions in Fig. 3.

# H   PRACTICAL APPLICATIONS OF META-CAUSAL MODELS

As the main focus of our initial work on meta-causal model lies on providing a first, formal, definition of meta-causal models, we tried to approach MCM from a spectrum of different theoretical

| Class Deviation | 1 Mechanism | 2 Mechanisms | 3 Mechanisms | 4 Mechanisms |
|:---:|:---:|:---:|:---:|:---:|
| 0.0 | 0.0349\|0.0590 | 0.0370\|0.0543 | 0.0380\|0.0592 | 0.0389\|0.0516 |
| 0.1 | " | 0.0381\|0.0555 | 0.0381\|0.0566 | 0.0346\|0.0528 |
| 0.2 | " | 0.0414\|0.0548 | 0.0412\|0.0567 | 0.0402\|0.0570 |

Table 4: **Mean average error for the slope and intercept of the correctly predicted mechanism for different class imbalances and number of mechanisms.** Mechanisms are accepted if their parameters do not differ by more than $0.2$ from the ground truth parameterization. The results show that converged samples are typically estimated with an error well below the threshold.

perspectives. In this Section, we provide a brief discussion on two possible practical fields of applications for MCM.

**Health and Medicine.** First, we would like to expand on our stress-induced fatigue example as it can not be reduced to a standard SCM, and provide an actionable perspective on MCM. While we still assume the underlying neuropsychological process to be more complex, with multiple interplaying factors to influence each other, we consider the same simplified model as presented in the paper. We now assume that some drug exists that is able to influence certain health related processes within the patient, such that the underlying –previously self-reinforcing– stress relation is unconditionally changed to a suppressing one. (Upon closer consideration, the previous intervention might constitute a meta-causal do-operator, as we detach the functional type from the underlying dynamics and fix its particular functional type.)

This perspective not only allows the forecast of system changes, but also yields an actionable model which can be actively steered between meta-causal states. To permanently treat a patient, one could consider the objective to reach a self-stabilizing meta-causal state. Note how this meta-causal objective might be different to that of a classical causal one, where stress levels would similarly be reduced, but no attention is placed on the (possibly unchanged) system dynamics, such that stress levels might rise up again after the intervention ends.

**Economics.** Second, recall that our MCMs are defined as finite state machines. Figuring out exact transition conditions that induce meta-causal state transitions also yield important insights on the volatility/stability of systems in terms of risk analysis and policy making. Such scenarios might commonly arise in economics, where relations in markets can change due to the sudden appearance of disrupting factors (e.g. a new competitor entering the market or a financial crisis) while effects might persist even with the disrupting factor having vanished.

## I   ASSERTIVE META-CAUSAL MODELS

In this section we will briefly touch upon possible assertive properties of meta-causal models. In this initial paper we chose to utilize a very general definition of types, which intentionally held flexible to allow for the most descriptive models. As for this definition, there might exist a gap between between the descriptive modeling of MCM and their 'assertive', data generating, properties. This gap primarily stems from the mapping of specific structural equations onto abstract types, which prevents a back projection of types to structural equations in the general case. With the special class of conditionally reducible SCM we, however, presented a particular class of MCM that yield 'assertive' properties by being able to translate types back onto the level of structural equations. Further restrictions on types and their relations to structural equations might be placed in specific applications, to shield users from misuse of the framework. Still, we where able to present several relevant applications of MCM even with this this most general formalization of MCM.

While meta-causal states might be mapped back to configurations of 'classical' causal models, we highlight that MCM are primarily concerned with the modeling of the overarching meta-causal state transitions. By modeling MCM as (possibly non-deterministic) finite-state machines, we, in fact, can make predictions about a systems' future course on the meta-level. This includes the sampling of state trajectories and making assertions about their stability and similar properties.

---

**Algorithm 1** Recovering Mechanisms for the Bivariate Case

---

1: **procedure** RECOVERMECHANISMS($\mathbf{x}, \mathbf{y}$, maxClassDev $K_{\max}$, EMSteps)
2:     **for all** $k \in [1..K_{\max}]$ **do**
3:         bestModelLogL $\leftarrow -\infty$
4:         N $\leftarrow$ requiredSamples($k$, maxClassDev, 0.95)     ▷ Compute # of samples (c.f. Sec. E)
5:         **for all** $n \in [1..N]$ **do**                   ▷ RANSAC iteration.
6:             $\mathbf{px}, \mathbf{py} \leftarrow \texttt{sample}(x_i, y_i, 2*k)$     ▷ Initialize parameters with $2 \times k$ points.
7:             **for all** $k' \in [0..k)$ **do**
8:                 $\alpha_k \leftarrow (py_{2k+1} - py_{2k})/(px_{2k+1} - px_{2k})$
9:                 $\beta_k \leftarrow py_{2k} - \alpha_k * x_{2k}$
10:                 $b_k \leftarrow 1.0$     ▷ Assume initial avg. deviation of the Laplacion to be 1.
11:                 $d_k \leftarrow$ 'XY'                 ▷ Assume $X \to Y$ direction first.
12:                 $\mathbf{c}_k \leftarrow P_{\texttt{Laplacian}}(\mathbf{x}, \mathbf{y}; \alpha, \beta, b, d)$     ▷ Initial class probabilities for all samples.
13:             **end for**
14:             **for all** $l \in [1..\text{EMSteps}]$ **do**                 ▷ EM Iteration.
15:                 $\boldsymbol{\alpha}, \boldsymbol{\beta}, \mathbf{b}, \mathbf{d} \leftarrow \texttt{regressLines}(\mathbf{x}, \mathbf{y}; \mathbf{c})$     ▷ (Weighted) median regression.
16:                 $\mathbf{c} \leftarrow P_{\texttt{Laplacian}}(\mathbf{x}, \mathbf{y}; \boldsymbol{\alpha}, \boldsymbol{\beta}, \mathbf{b}, \mathbf{d})$
17:             **end for**
18:             modelLogL $\leftarrow \sum_i \text{LogP}_{\texttt{Laplacian}}(\mathbf{x}, \mathbf{y}; \boldsymbol{\alpha}, \boldsymbol{\beta}, \mathbf{b}, \mathbf{d})_i$     ▷ Obtain the joint log probs.
19:             **if** modelLogL $>$ bestModelLogL **then**
20:                 bestModelLogL $\leftarrow$ modelLogL
21:                 bestParameters $\leftarrow (\boldsymbol{\alpha}, \boldsymbol{\beta}, \mathbf{b}, \mathbf{d})$
22:             **end if**
23:         **end for**
24:         allLaplacian $\leftarrow$ `true`                 ▷ Check if residuals are Laplacian distributed.
25:         **for all** $l \in [1..k]$ **do**
26:             $(\boldsymbol{\alpha}, \boldsymbol{\beta}, \mathbf{b}, \mathbf{d}) \leftarrow$ bestParameters
27:             $\mathbf{x}^l, \mathbf{y}^l, \mathbf{c}^l \leftarrow \texttt{selectClass}(\mathbf{x}, \mathbf{y}, \mathbf{c}, l)$     ▷ Select points where argmax$_{\mathbf{c}} = l$.
28:             $\mathbf{x}^l, \mathbf{y}^l \leftarrow \texttt{filter}(\mathbf{x}^l, \mathbf{y}^l, \mathbf{c}^l, 0.4)$ ▷ Keep points s.t. $\max^{\#2}\mathbf{c}^l < 0.4(1 - \max^{\#1}(\mathbf{c}^l))$.
29:             $r = \texttt{LinEq}(\mathbf{x}^l, \mathbf{y}^l; \alpha_l, \beta_l, d_l) - \mathbf{y}^l$
30:             **if not** $\texttt{AndersonDarling}(r, 0.95)$ **then**
31:                 allLaplacian $\leftarrow$ `false`
32:                 **break**
33:             **end if**
34:         **end for**
35:         **if** allLaplacian **then**
36:             **return** k
37:         **end if**
38:     **end for**
39:     **return** 0
40: **end procedure**

---

