# OpenReview forum: "Systems with Switching Causal Relations: A Meta-Causal Perspective"
_ICLR.cc/2025/Conference — ICLR 2025 Spotlight_

### Official Review · Reviewer_puHX · 2024-10-30

**Soundness:** 3
**Presentation:** 3
**Contribution:** 4
**Rating:** 8
**Confidence:** 2

**Summary:**

The paper introduces the concept of meta-causal models to formalize the emergence of causal relationships, which, for instance, allows one to formalize changes in the causal structure over time. While more classical approaches based on graphical causal models assume fixed causal relationships, the proposed meta-causal models can represent and reason about situations where causal mechanisms emerge, change, or disappear. The authors formalize meta-causal states that group similar causal behaviors and demonstrate how to infer these states from data in some toy scenarios.

**Strengths:**

+ Novel theoretical contribution towards a more generalized way to formalize causality
+ Mostly clear motivation, although the introduction could be improved
+ Although limited, the authors aimed to not only provide a theoretical discussion, but also insights towards a practical application
+ Fair discussion of the limitation of the work

**Weaknesses:**

- While certain aspects of the theory are well introduced, theoretical (proven) statements and formalization of axioms are somewhat lacking, considering that the goal of the work is to establish a new formalism
- The experimental evaluation is somewhat limited, but this is a minor point as the theoretical contribution is the main focus
- The introduction is too complex and abstract
- The practical applicability to larger, more complex systems remains unclear

**Questions:**

Overall, the paper provides a great first step towards an alternative formalism to describe and formalize the emergence of causal relationships. Since such types of works typically are more focused on the theoretical work and introduction of new formalisms, the concerns about the practical and experimental aspects are rather minor.

Generally, I would have expected a few more discussions related to the distinction between meta-causal states and traditional latent variable approaches. Although I am not that familiar with related work in this domain, I would have expected some more discussions with work introducing alternative ways to formalize causality, such as:
- "A Measure-Theoretic Axiomatisation of Causality" by Park et al.
- "Phenomenological Causality" by Janzing et al.

These works go in a different direction but seem, on a high level, related.

The introduction is very hard to follow and could be motivated better. While the example in Figure 1 is definitely helpful to understand the idea and limitations, the introduction often sounds too abstract. Maybe starting with the Figure 1 (a) example on a high-level description without further introduction of any formalisms around causal modeling could be helpful for the motivation.

While the practical application in a general setting seems to be limited in the current form, the authors fairly discuss the current limitations and the work still provides a valuable foundation for follow-up work.

When the authors can comment on the lack of theoretical statements (e.g., theorems ensuring that, under certain conditions, one can transform a MCM to a classical SCM), I am willing to further increase the score.

---

> ### Author Response · Authors · 2024-11-15
>
> Dear Reviewer puHX,
> Thank you for your thoughtful review and for highlighting our strengths, including the novel theoretical contributions, clear motivation and insight for practical application. We will address your questions in the following:
>
> **Q1 [Relation to Measure-Theoretic Axiomatisations and Phenomenological Causality]:**
> Thanks for referencing both works to us. We cite and (due to space constraints) briefly discuss the relation of measure-theoretic considerations towards the induction of causal relations in mediation processes, as well as the role of phenomenological considerations towards the overall constitution of causal variables.
>
> **Q2 [Abstract Motivation / Practical Applications]:** [“The introduction is very hard to follow and could be motivated better. While the example in Figure 1 is definitely helpful to understand the idea and limitations, the introduction often sounds too abstract. Maybe starting with the Figure 1 (a) example on a high-level description without further introduction of any formalisms around causal modeling could be helpful for the motivation.]
>
> Since we are first to introduce a rather novel formal definition of meta-causality within the restricted space of a conference paper, we are forced to keep things compact. Examples of meta-causality, by their nature, presume some existing understanding of classical causal models. Here, we decided to trade the inclusion of multiple relevant perspectives and applications of meta-causality in the main paper (identifiability of meta-causal states [game of tag], relation to contextual independencies, attribution of policy, discovery of meta-causal states and the stress induced fatigue example) against a short and, admittedly, rather technical introduction.
>
> **Q3 [Lack of Theoretical Statements]:**
> The role of contextual independencies with regard to the reduction of MCM onto SCM was discussed in section 3, but indeed lacked a rigorous theoretical foundation.
>
> After further consideration of the problem we are happy to present the conditions under which MCM can be reduced to SCM with conditioned structural equations. Roughly speaking, a reduction to SCM is possible whenever all transitions on the meta-causal model are self-loops. Generally, this is the case when the types of structural equations are determined by their functional form and do not change dynamically with the current state of the system.. Based on your feedback and requests of the other reviewers, we provide and prove a theorem for the exact conditions for the reduction of MCM. Due to space constraints the theorem and proof has been placed in Appendix C “Reduction of Meta-Causal Models to Conditioned SCM”.
>
> ------------------
>
> We have uploaded a revised version of the paper, with changes color coded, for which we hope to have clarified most of your stated concerns. Your color code is **orange.** Please let us know if there is any additional information we can provide to further clarify our contributions. Thank you once again for your time and effort in reviewing our paper!
>
> Sincerely,
> Authors

---

> > ### Comment · Reviewer_puHX · 2024-11-19
> >
> > Thanks a lot for your response and careful revision. I am increasing my score.

---

### Official Review · Reviewer_PEUy · 2024-11-02

**Soundness:** 4
**Presentation:** 3
**Contribution:** 2
**Rating:** 8
**Confidence:** 2

**Summary:**

The manuscript introduces a framework to analyze dynamic causal relationships that shift based on environmental or internal changes, called meta-causal models (MCM). The framework extends traditional causal models by defining “meta-causal states,” which group standard causal graphs into clusters of qualitative behavior. Then, MCM can dictate the detection of underlying changes in causal structures.

**Strengths:**

* The manuscript provides a formal framework for handling the dynamics of an SCM. Definitions 1 to 3 are elegant implementations of these ideas. In particular, framing meta-causal models as finite state machines is intuitive and well-suited for applications.
* The framework allows for a systematic attributing responsibility in dynamic scenarios. In Section 4.1, the manuscript shows how counterfactual scenarios can be better explained in a meta-causal model. Furthermore, the paper provides a clear example of when attributing effects via model parameters might fail to explain preventative mechanisms.

**Weaknesses:**

* Meta-causality scope: the paper isn’t clear on two philosophical aspects of the framework regarding its scope and interpretation, namely (i) the causal-effect relationship between the meta-causal model and the causal model instance it is working with and (ii) the potential sutil difference between meta causal edges and classical ones
	- (i) From the initial example in Figure 1, the reader could question if the meta-causal model is causing the underlying classical causal model. This discussion isn’t handled in the manuscript, but there is an acknowledgment note in Section 3 under “Causal and Anti-Causal Meta-Causal States.”
	- (ii) How do we interpret and handle the subtle difference between meta-causal and classical relationships? For instance, in Fig 1, a classical causal discovery algorithm could attribute Api and Bx as causes of Ax.

**Questions:**

* Are there other examples or mechanisms where classical models will fail to capture via observed changes in variable values and meta-causal models won’t? How often or realistic are the preventive mechanisms, like the ones described in Section 4.1?

---

> ### Author Response · Authors · 2024-11-15
>
> Dear Reviewer PEUy,
> Thank you for your thoughtful review and for highlighting our strengths, including the elegant implementation of SCM dynamics, the framing of meta-causal models as finite state machines, and the application of meta-causal attributions for preventive mechanisms. We will address your concerns in the following:
>
> **W1: [Do Meta-Causal Model Cause Standard Causal Model?]**
> Thank you for raising this important question, we will try to clarify this in the following. Please note that, upon the feedback of you and reviewer DQ6p we have revised the definition of meta-causal frames in Section 3 to make the relation between the mediation process, MCM and SCM more clear and provide an extended discussion in the appendix. We believe that our answer Q3 to DQ6p might also be insightful in that regard.
>
> Meta-Causal Models do not cause SCM, but are descriptive models that model how changes in the underlying process are reflected in the causal graph and, therefore, capture the changing dynamics of SCM. To be more specific, consider that the type encoder derives the type of edges based on the functional relationship between variables $X_j$ on $X_i$ by considering the underlying mediation process. The MCM is governed by the mediation process, as is the SCM. However, both models pursue a different perspective on the process. While SCM are concerned with how values propagate through the system, MCM are concerned how system dynamics affect the causal graph.
>
> **W2: [Subtle Differences Between Standard and Meta Causal Models.]**
> You are correct in pointing out that meta-causal models might be reduced to classical SCM under certain conditions. The role of contextual independencies with regard to the reduction of MCM onto SCM was already discussed in section 3, but indeed lacked a rigorous theoretical foundation. As also requested by other reviewers, we are happy to extend our formalism and prove conditions under which this reduction is possible in Appendix C “Reduction of Meta-Causal Models to Conditioned SCM”. Roughly speaking, a reduction to SCM is possible whenever all transitions on the meta-causal model are self-loops. Generally, this is the case when the types of structural equations are determined by their functional form and do not change dynamically with the current state of the system. Note in that regard, that the Stress-Induced Fatigue example (Sec. 4.3) provides a counter-example where a reduction to a conditioned SCM is not possible for the reasons discussed in the paper. In such scenarios meta-causal inferences might be able to identify general conditions that induce changes in the SCM, while classical causal inferences are bound to reason within the currently given causal graph.
>
> **Q1: [Other examples where classical examples fail. How realistic are preventive mechanisms?]**
> We assume that preventive mechanisms might be more common than expected. Consider for example the implications of the mental jenga example in Zhou et al. [1]. Even though the initial stack of blocks is static, --in the sense that no block is falling without external perturbation, therefore *preventing* the actual effect of gravity--, it is clear that the removal of any of the blocks causes others to fall. While removing individual blocks (with idealized precision) from a jenga tower might initially not yield apparent changes, it is clear that not all blocks of a row can be removed. In that sense preventive mechanisms might be encountered in every-day life as humans consider them as cognitive abstractions of the structural mechanics that govern our every-day environment.
>
> [1] Zhou, Liang, et al. "Mental jenga: A counterfactual simulation model of causal judgments about physical support." *Journal of Experimental Psychology: General* 152.8 (2023): 2237.
>
> ------------------
>
> We have uploaded a revised version of the paper, with changes color coded, for which we hope to have clarified most of your stated concerns. Your color code is **green.** Please let us know if there is any additional information we can provide to further clarify our contributions. Thank you once again for your time and effort in reviewing our paper!
>
> Sincerely,
> Authors

---

> > ### Author Response · Authors · 2024-11-27
> > **Any further questions?**
> >
> > Dear Reviewer,
> >
> > We hope to have clarified all your concerns. Let us know if there are any further concerns. If no, it will be great if you could reconsider your score.
> >
> > Regards,
> >
> > The Authors

---

> ### Comment · Reviewer_PEUy · 2024-11-27
> **Scoring update**
>
> I thank the authors for addressing my concerns and providing the updated paper. Appendix C, "Reduction of Meta-Causal Models to Conditioned SCM," strengthens the manuscript. I'll update my recommendation accordingly.

---

### Official Review · Reviewer_DQ6p · 2024-11-04

**Soundness:** 2
**Presentation:** 2
**Contribution:** 3
**Rating:** 6
**Confidence:** 3

**Summary:**

This paper defines meta-causal systems to allow us to ask what caused a structural equation to be the way it is. This allows us to attribute responsibility for some variable sometimes to another variable, and sometimes to an external force that intervened on the system, as appropriate. An algorithm using RANSAC and EM is proposed to learn such meta-causal systems from data.

**Strengths:**

The idea of looking for the causes behind an SCM is very interesting and could be relevant in certain contexts. Putting these ideas in practice required the development of a nontrivial algorithm; while generalizing it is left for future work, this does help to solidify the contribution.

**Weaknesses:**

- The core concept of the paper seems to be multiply defined. I understand "meta-causal" to refer to questions of what *caused* the structural equations to be what they are (as in line 044-045). But in other places in the paper, it seems to refer to an abstraction where we don't look at the precise structural equations but instead at some qualitative characteristics. E.g. in the dynamical system of stress-induced fatigue in section 4.3, the structural equations remain the same but it is emphasized that there are are different causal states (e.g. self-reinforcing or self-suppressing). It is presented as a major premise of the paper (final sentence of abstract) that meta-causal states do not always result from external influences. But according to the first definition, there is only one meta-causal state, because the structural equations don't change, so this is not an example of meta-causality. (As a consequence of this ambiguity of "meta-causality" in the paper, I find it hard to assess to what extent the emprical results support the claims of the paper.)
- I found the start of section 3 to be rather hard to follow, due to its density and it being unclear what intuition is being formalized. There is no overall description of how SCM and mediation process make up a more complex new process. The transition function seems to imply a dynamical view, with discrete time steps. But it is not made explicit anywhere that variables should be thought of as having a time index (should they?). In definition 1, the presence of the transition function does suggest that transitions between time steps are always happening.

**Questions:**

- In section 3, I assumed from the matching notation that the variables $\mathbf{X}$ are part of an SCM. Are they?
- In definition 1, line 141, in what way does an element of  $\mathcal{X}_j^{S}$ represent a "change" of $\mathcal{X}_j$?
- An odd point I wanted to note: if an SCM has arrows $X \to Y$ and $Y \to Z$, the meta-causal state would also have an arrow for the indirect effect $X \to Z$. So these arrows aren't structural, merely descriptive. It depend on the answer to my question about $\mathbf{X}$ above whether this is a drawback or merely remarkable.
- Line 171 "As standard causal relations emerge from the underlying mediation process": How do they emerge? Is this always true?

---

> ### Author Response · Authors · 2024-11-15
>
> Dear Reviewer DQ6p,
> Thank you for your thoughtful review and highlighting the interesting and relevant ideas, and the development of a nontrivial algorithm evaluation. We will address your concerns in the following:
>
> **W1 [What is “Meta-Causal”]**
> We are sorry for the confusion regarding the use of meta-causality throughout the paper. In particular, and as identified by other reviewers, meta-causality is concerned with the underlying process dynamics that govern the structural equations of an SCM. This can indeed sometimes be reduced to the pure consideration of structural equations conditioned on some external latent factor Z. In particular, the initially provided example falls into this class of models. Note that these models however only exhibit transition functions with only self-loops and feature no transitions between states (c.f. discussion on the “Role of Contextual Independencies” at the end of Sec. 3). As MCM are also able to model systems that can transition between meta-causal states (e.g. change relations of the causal graph in relation to changes in the underlying process), MCM form a strict superset of the class of latent conditioned SCM. As you pointed out correctly, we show such a case in the stress-induced fatigue example, where the system system dynamics --the types of the edges-- also depend on the current state of the system, and therefore can not be solely controlled via some external factor $Z$.
>
> **Q1 [Variables $\mathbf{X}$]**
> Yes, variables $\mathbf{X}$ are the variables of the SCM. We have clarified this in the paper.
>
> **Q2 [Definition of the Type Encoder]**
> Thank you for pointing out this weakness! After reconsidering the definition of our type encoder, we decided to revise said definition to make section 3 more approachable and better defined in general. We are happy to explain further: The type encoder assigns types based on the functional relationship between variables $X_j$ on $X_i$ as induced by the underlying mediation process. In particular the type encoder considers the projection of the transition function onto $X_j$, which we now define as $\varphi_j \circ \sigma$. The type encoder, therefore, is the key binding factor between the meta-causal model and the underlying mediation process. In particular, $\tau_{ij}$ reasons about the relationship between variables $X_j$ and $X_i$ by inspecting properties of $\varphi_j \circ \sigma$.
>
> Consider, in the stress induced fatigue example (Sec. 4.3), we employed an identification function to determine the functional relation of stress $s$ onto itself via the derivative $d(\varphi_s \circ \sigma) / ds$ to be either of reinforcing or suppressing nature. Similar to our next answer (Q3), where indirect edges may constitute a type, general classes of functional relationships, e.g. linear functions could be identified as independent types.
>
> **Q3 [Inclusion of Indirect Edges]:** Whether to include or not indirect edges is indeed an interesting question and depends on the objective of the user, and ultimately on the deployed identification function. The answer eventually reduces to the question of whether indirect effects should be considered causal effects. Under the assumption that the intermediate variable $Y$ is not part of the SCM we surely want our identification function to include $X \rightarrow Z$, while whenever $Y$ is part of the SCM we rather want to include $X \rightarrow Y, Y \rightarrow Z$ and omit the indirect edge.
>
> One could (and, in general, probably should) define the identification function to only identify direct causal effects. Given our meta-causal framework we, however, decided to leave this freedom to the practitioner. In this case one might assign particular “1-hop indirection” types to the edges - indicating the edge to be the result of an indirect relation via some (possibly unobserved) variable. Consider how this notion of edge types over unobserved variables is already in use in CPDAGs (as e.g. produced by the PC or GES algorithms) where it is used to indicate undirected or confounded edges.
>
> [*our reply is continued in the next comment*]

---

> > ### Author Response · Authors · 2024-11-15
> >
> > **Q4 [Role of Time]:** Within our definition of meta-causal frames (Def. 1), we ground the emergence of SCM to an underlying mediating process. In particular, for any given meta-causal frame, particular causal interactions are identified via the identification function. While the underlying mediation process is time-dependent, the resulting causal graph might be a time-independent projection of all causal interactions within the process onto a graphical structure. Note that the resulting SCM still preserves the underlying sequence of causal interactions through the DAG induced partial ordering of variables. This ‘logical’ time ordering (induced via the partial order) can be seen as an abstraction of the underlying process. Under this perspective, meta-causal frames are important from a theoretical perspective as they justify the conditions under which SCM emerge and thus, under which conditions they are expected to change.
> >
> > We have added all of the above discussions to the appendix and refer to them at the start of Section 3. We hope these answers and our reply to Q2 also resolve W2 to some extent by clarifying the relationship between mediating processes, the emergence of SCM and the relation to meta-causal models.
> >
> > ---
> >
> > **Polite Request for Re-evaluation**
> >
> > Dear Reviewer, Thank you once again for your time and effort in reviewing our paper and in particular for pointing out ambiguities that helped us improve our formalization of MCM. We have uploaded a revised version of the paper, with changes color coded, for which we hope to have clarified most of your stated concerns. Your color code is **red.** Please let us know if there is any additional information we can provide to further clarify our contributions.
> >
> > Sincerely,
> > Authors

---

> > > ### Author Response · Authors · 2024-11-25
> > > **Any further questions?**
> > >
> > > Dear Reviewer,
> > >
> > > We hope to have clarified all your concerns. Let us know if there are any further concerns. If no, it will be great if you could reconsider your score.
> > >
> > > Regards,
> > >
> > > The Authors

---

> > > > ### Comment · Reviewer_DQ6p · 2024-11-27
> > > >
> > > > Dear authors, thank you for your replies which help make several aspects of your paper more clear. However, there are some points that still confuse me. I hope you can clarify these further.
> > > >
> > > > - **W1 \[What is meta-causality\]:** Could you please provide a crisp, unambiguous definition of what you mean by "meta-causality"? In your previous answer, you write: "meta-causality is concerned with the underlying process dynamics that govern the structural equations of an SCM". If this is to be taken as a definition of meta-causality, I find it surprising that when the structural equations are not changing (such as in the stress-induced fatigue example), it is still considered in the paper as an example of meta-causality. It seems to me that the definition would somehow need to incorporate the fact that instead of the structural equations, certain *qualitative* aspects of the process must be considered.
> > > >
> > > > - **Q3 \[Inclusion of Indirect Edges\]:** Your reply/Appendix B, if I read them correctly, suggest that we have the freedom to choose a different identification function depending on how we want indirect causal relations to be treated. However, the identification function is fully specified in definition 1; the only freedom this definition allows is in the choice of type encoder. But even then, both identification function and type encoder are functions only of two variables, and can't change their output depending on whether or not a mediating variable exists between them. So my conclusion remains that definition 1 is unsuitable to cases involving indirect causal relations. (BTW, as a correction to the final sentence of that appendix: the undirected edges in CPDAGs represent direct causal relations of unknown direction.)
> > > >
> > > >
> > > > ### Other remarks
> > > > (These are remarks on the original submission that apparently I forgot to copy to OpenReview. Some of the mentioned texts might have moved or changed already in the revised manuscript, but I hope overall they are still helpful.)
> > > > - l103: "for all variables" -> "for each variable"
> > > > - l117 talks about a set of functions, but in section 3 there is one function with a non-Boolean codomain
> > > > - l184: "disjunct" -> "disjoint"
> > > > - equation in line 200: The right-hand side equation expresses that A is giving chase. But the arrow $A \to B$ on the left refers to the structural equation that describes *B*'s behavior!
> > > > - "Role of Contextual Dependencies": I suppose the word "concatenation" on line 234 refers to the function composition just before the equation. (An index is wrong there.) I think more explanation is needed of how the $f'$ relate to the $f$. If I understand the intention correctly, there is still the problem that the domains and codomains of the $f'$ might not be compatible in such a way that we can just remove a function from the composition.
> > > > - footnote 2 on page 7: "both direction" - but in the experiment, the two arrows can't both be present at the same time, right?
> > > > - There are relatively many typos on the final pages (8-).

---

> ### Author Response · Authors · 2024-11-27
>
> Dear Reviewer,
> thank you for being back and engaging in the discussion of our paper. Considering your further comments we reconsidered parts of our paper and fixed the mentioned inconsistencies in our formalism. We detail the changes below:
>
>
>
> **W1 [Meta-Causality]:** In your quoted sentence we regard the qualitative behavior of equations to be part of the "process dynamics". You are right, however, that talking about "structural equations" in this context is an incorrect phrasing, since --as you mention correctly--, the same structural equations can exhibit different qualitative behavior. We checked our paper, and found that --while avoiding it elsewhere-- we accidentally introduced this phrasing in our added clarification in the introduction.
>
> Just as there are different definitions of causality, the 'exact' definition of meta-causality will likely depend on the author. However, as we are (probably) first to give such a definition, we orient ourselves on previous definitions of causality. Our paper currently defines types as the "general qualitative behavior that emerges from the interplay of individual equations" and we would like to find a compatible definition for meta-causality. If we consider causality to be "the science of cause and effect", as prominently stated by the Book of Why, meta-causality might be defined as "[the science of] qualitative change in cause-effect behavior".
>
> We have adjusted the previously added sentence in the introduction and use this new definition at the start of section 3. Thank you again for challenging us to reduce our rather vague definition of meta-causality and provide a more condensed version. We would be interested on whether you agree with us on this definition or not.
>
> **Q3 [Indirect Edges and Identification Function]:** You are right in that our presented formalism, regarding the type encoder (and therefore the identification function), was somehow not properly formalized. Previously, we considered the type-encoder as a part of a Meta-Causal Frame, and therefore to be implicitly be aware the mediation process via its definition. However, in that case it is rather confusing to pass the particular mechanism $X^{\mathcal{S}}_j$ of the variable $X_j$ as an explicit parameter, as it should already be contained in the overall known set of mechanisms $X^{\mathcal{S}}$.
>
> To resolve this inconsistency, we drop the subscript in $\mathcal{X}^S_j$ and explicitly pass all functional relations $\mathcal{X}^S$ to the identification function and the type encoder. In particular, this allows the type decoder to determine whether some a functional relation is mediated by some other (set of) intermediate mechanism(s). In situations where $X_i \rightarrow X_{\mathbf{z}} \rightarrow X_j$, the type encoder needs to identify whether the relation $X_i \rightarrow X_j$ is purely mediated via $\mathcal{X}^S_{\mathbf{z}}$ (meaning that all computational paths for $X_i \rightarrow X_j$ pass through $\mathcal{X}^S_{\mathbf{z}}$ at some point) or some additional direct effect exists that are not 'shielded' by $\mathcal{X}^S_{\mathbf{z}}$. Conversely, whenever some $X_k \in X_\mathbf{z}$ of the intermediate set is dropped or marginalized from the causal variables, the mechanism $\mathcal{X}^S_{k} \subset \mathcal{X}^S_j$ is no longer identified as a subcomponent of the mechanism within the overall $\mathcal{X}^S_j$ and (given that the remaining $\mathcal{X}^S_{\mathbf{z}\setminus \{ k \}}$ might no longer shield $X_i$ from $X_j$) a direct arrow $X_i \rightarrow X_j$ can be inferred.
>
> We adjusted definition 1 and carried the corresponding changes to the text and notation. Additionally, we added this discussion, regarding the relation of all (and possibly intermediate) mechanisms $X^{\mathcal{S}}$ and the role of the specific $X^{\mathcal{S}}_j$, to Appendix B. (We corrected the statement regarding CPDAGs. Thanks!)
>
> **[Game of Tag Example]:** Thanks for pointing this out. We corrected the right-hand side of the equation to state $t_{B \rightarrow A} = \text{``A Chasing''}$ to now indicate the correct edge.
>
> *[continued in the next comment]*

---

> ### Author Response · Authors · 2024-11-27
>
> **[Contextual Independencies]:** Yes. We, indeed, meant function composition in the mentioned case (and have adjusted the wording). Based on your feedback and that of the other reviewers, we already revised the corresponding paragraph and equation and [have added the following clarifications to the current revision (Appendix B)]:
>
> Since we construct the conditioned SCM after the particular causal model that exists under a meta-causal state indexed by $z$, we know that a particular composition of functions $f^z_{ij}$ has to exist (and that the functions are compatible), since the full function $f^z_j$ exists for each meta-causal state. Consider that, in a naive approach, the order of composition $i \in \{1..N\}$ enforces a particular evaluation order of the functions, and in particular requires this order to be the same for every meta-causal state.
>
> Generally, the presented function composition might not work, in the case that the individual functions get chosen badly from the start. A simple solution to overcome such problem is to assume compositions of lifted functions and assume their signatures to be compatible (which is always permitted due to the known existence of the composed $f^z_j$). Note that functions $f^z_{ij}, f^{z'}_{ij}$, *[new paragraph due to openreview latex formating issues, sorry!]*
>
> whose functional type $t_{i j}$ did not change under a change of $z$ to $z'$, might be altered to make signatures compatible.
>
> In the paper we now explicitly state compatibility of the functions and have added this extended discussion to the appendix C.
>
>
>
> **[Experiments: Causal Effect Direction]:** In our experiments we exclusively consider either $X\rightarrow Y$ or $Y \rightarrow X$. However, having a cyclic relation will likewise realize a particular distribution. We assume that this case might still be identifiable as the noise distribution might no longer be Laplacian distributed from either direction, due to the interfering noise from both equations.
>
>
>
> **[Typos]** Thank you for pointing out the specific typos in our paper. We have them fixed. We will perform an more thorough correction pass for the whole paper and in particular pages 8 and following for the camera-ready version.
>
>
>
> ---
>
> Thank you once again for your detailed comments and suggestions regarding our paper. We have included all discussed points in the current revision of the paper. While we might not be able to upload further revisions after end-of-day AoE due to the current ICLR policy, we are happy to discuss and clarify any remaining points that might have been left unclear.

---

> > ### Author Response · Authors · 2024-11-30
> > **Any further questions?**
> >
> > Dear Reviewer,
> >
> > We hope to have answered your follow up questions in detail. Let's us know if your concerns are resolved. We are looking forward to hearing your thoughts.
> >
> > Regards,
> >
> > The Authors

---

> > > ### Comment · Reviewer_DQ6p · 2024-12-02
> > >
> > > Dear authors,
> > >
> > > Thank you for your response. I'm glad to see your clarified definition of "meta-causality".
> > >
> > > I have some concerns about the changes to how type encoders work (referring to the changes in the revised manuscript as well as those described in your reply). With these changes, for any SCM, one can tailor a different type encoder to each pair of variables, and have it choose a "type" depending on, it seems, basically any property of the SCM's variables. This flexibility makes the notion of "type" very vague. Also, while I think this additional flexibility is intended to allow distinguishing direct from indirect causal relations, in fact the type encoding is still *descriptive* of the SCMs behaviour (as opposed to *defining* its behaviour, like how structural equations define the values of variables in different imaginable circumstances). I remarked the same about the original definition. Wouldn't it then be better to revert to the original definition, but admit that it is descriptive only?

---

> > > > ### Author Response · Authors · 2024-12-02
> > > >
> > > > Dear reviewer,
> > > > indeed, we chose the definition of types intentionally flexible as opposed to having to place rather conservative restrictions to shield users from misuse of our framework. As with most things, if a user intends to operate our framework incorrectly, they won't yield meaningful results... On the other side, we have presented several clearly relevant applications of MCM in our paper.
> > > >
> > > > We agree that there might currently exist a gap between between the descriptive modeling of MCM and their 'assertive', data generating, properties. This gap primarily stems from the mapping of specific structural equations onto abstract types, which prevents a back projection of types to structural equations in the general case. With the special class of conditionally reducible SCM we have however presented a particular class of MCM that yield this 'assertive' property by being able to translate types back onto the level of structural equations.
> > > >
> > > > Having said all this, consider that our MCM are primarily concerned with the modeling of the overarching meta-causal state transitions. By modeling MCM as (possibly non-deterministic) finite-state machines, we, in fact, can make predictions about a systems' future course on the meta-level. This includes the sampling of state trajectories and making assertions about their stability and similar properties.
> > > >
> > > > We will provide a reference to this more nuanced perspective on the assertive properties of MCM in the conclusion section of our paper.

---

> > > > > ### Author Response · Authors · 2024-12-03
> > > > > **Rebuttal phase closing**
> > > > >
> > > > > Dear Reviewer,
> > > > >
> > > > > Thank you again for the fruitful discussion and engagement.
> > > > > We hope we have clarified your concerns. Looking forward to hearing from you.
> > > > >
> > > > > Regards,
> > > > >
> > > > > Authors

---

### Official Review · Reviewer_pnp9 · 2024-11-05

**Soundness:** 4
**Presentation:** 3
**Contribution:** 3
**Rating:** 8
**Confidence:** 3

**Summary:**

The paper introduces the concept of *meta-causal states*, which attempt to represent how the underlying dynamics of a causal model can change based on context. The authors then examine how meta-causal states could be used in the attribution of responsibility, describe how the number of distinct meta-causal states can be identified empirically (a variety of EM), and provide an example analysis of stress-induced fatigue.

**Strengths:**

The concept is interesting and the paper is well-written and detailed.

Section 4 presents some diverse and interesting applications of the basic theory.

The graphics, particularly Figure 1, are clear and easy to understand.

**Weaknesses:**

The paper should cite some relevant prior work in graphical models with “gates” (Minka 2008; Winn 2012). The concepts in those papers are somewhat different, but closely related to the topic of this paper.

The paper should note that some meta-causal states can actually be represented by ordinary SCMs. For example, in an SCM in which Z is caused by both X and Y, it is entirely possible for Y to control *the manner in which* X affects Z. In the simplest case, Y could be a binary variable that determines the functional form of how X affects Z. This complicates the structural equation of Z, but it’s not outside of the standard definition of SCMs. Of course, this approach has limitations. For example, it cannot be used to reverse the direction of causation between two other variables (at least not within the standard definition of SCMs). However, it can toggle the existence of an edge or change the manner in which one or more parent variables affects their child. This should be noted more clearly in the discussion.

The utility of meta-causal states is not immediately evident. That is, what sorts of practical causal inference in medicine, economics, social science, or other domains would be improved by consideration of meta-causal states? What errors are made now using conventional causal modeling that would be avoided by the use of meta-causal states? I suspect that there are good answers to these questions, but it would improve the paper if they were more clearly explained.

**References**

Minka, T., & Winn, J. (2008). Gates. *Advances in Neural Information Processing Systems*, *21*.

Winn, John. "Causality with gates." In *Artificial Intelligence and Statistics*, pp. 1314-1322. PMLR, 2012.

**Questions:**

What sorts of practical causal inference in medicine, economics, social science, or other domains would be improved by consideration of meta-causal states?

What errors are made now using conventional causal modeling that would be avoided by the use of meta-causal states?

---

> ### Author Response · Authors · 2024-11-15
>
> Dear Reviewer pnp9,
> Thank you for your thoughtful review and for considering our work to be well-written, providing interesting applications, and to provide clear and easy to understand graphs. We will address your concerns in the following:
>
> **W1 [Prior Work]**
> Thank you for referencing further relevant prior work that slipped our attention. We agree that the mentioned papers are closely related to the topic of modeling meta-causal function switching via context-independencies and have added a brief discussion to our paper.
>
> **W2 [Reduction of MCM to ordinary SCM]**
> We agree with your concerns in the lack of a formal theorem. As already stated in the paper and pointed out by the other reviewers PEUy and puHX, meta-SCM might --under certain conditions-- be reduced to ordinary SCM. Note that the Stress-Induced Fatigue example (Sec. 4.3) provides a counter-example where a reduction to a conditioned SCM is not possible for the reasons discussed in the paper. We are delighted to provide and have proven a theorem stating sufficient conditions for when a reduction of MCM to standard SCM is possible. Roughly speaking, a reduction to SCM is possible whenever all transitions of the meta-causal model are self-loops. Generally, this is the case when the types of structural equations are determined by their functional form and do not change dynamically, based on the current (value) state of the system. We have revised Equation 1 and due to space constraints, provide the corresponding theorem and its proof in Appendix C.
>
> **Questions/W3 [Utility of Meta SCM / Errors of inferences of conventional causal models]**
> As the main focus of our initial work on meta-causal models lies in providing a first formal definition of meta-causal models, we tried to approach MCM from a spectrum of different theoretical perspectives. As a response to your feedback, we will discuss possible practical applications in the following and have added this discussion in Appendix G to the paper.
>
> Regarding possible errors of conventional causal model inference, consider the motivational example where classical and meta-causal attributions come to different conclusions; with conventional inference stating that B was the root-cause of A following it. It is not so much that one or the other perspective is per se ‘incorrect’, but that meta-causal analysis allows reasoning over structure dynamics of a system, while conventional causal analyses might be bound to the respective current graph structure.
>
> **Health and Medicine.** We would like to expand on our stress-induced fatigue example as it can not be reduced to a standard SCM, and provide an actionable perspective on MCM. While we still assume the underlying neuropsychological process to be more complex, with multiple interplaying factors to influence each other, we consider the same simplified model as presented in the paper. We additionally assume that some drug exists that is able to influence certain health related processes within the patient, such that the underlying --previously self-reinforcing-- stress relation is unconditionally changed to a suppressing one. (Upon closer consideration, the previous intervention might constitute a meta-causal do-operator, as we detach the functional type from the underlying dynamics and fix its functional type.)
>
> This perspective not only allows the forecast of system changes, but also yields an actionable model which can be actively steered between meta-causal states. To permanently treat a patient, one could consider the objective to reach a self-stabilizing meta-causal state. Note how this meta-causal objective might be different to that of a classical causal one, where stress levels would similarly be reduced, but no attention is placed on the (possibly unchanged) system dynamics, such that stress levels might rise up again after the intervention ends.
>
> **Economics.** Recall that our MCMs are defined as finite state machines. Figuring out exact transition conditions that induce meta-state transitions also yield important insights on the volatility/stability of systems in terms of risk analysis and policy making. Such scenarios might commonly arise in economics, where relations in markets can change due to the sudden appearance of disrupting factors (e.g. a new competitor entering the market or a financial crisis) while effects might persist even with the disrupting factor having vanished.
>
> While we are only able to briefly discuss possible areas of application in this paper, we believe that applications of MCM are plentiful and are keen to expand applications in follow up works.
>
> ---
>
> We have uploaded a revised version of the paper, with changes color coded, for which we hope to have clarified most of your stated concerns. Your color code is **blue.** Please let us know if there is any additional information we can provide to further clarify our contributions. Thank you once again for your time and effort in reviewing our paper!
>
> Sincerely,
> Authors

---

### Meta-Review · Area_Chair_Pksi · 2024-12-23

**Metareview:**

This paper proposes meta-causal models (MCM) to handle changing causal structures, extending classical causal models so that the system’s “causal equations” can shift based on state or context. Reviewers praised the novelty of modeling how causal mechanisms themselves emerge and switch, appreciating the theoretical framework, finite-state-machine representation, and illustrative examples. Concerns centered on clarity—especially definitions of “meta-causality,” how “type encoders” handle direct vs. indirect effects, and when MCMs reduce to standard SCMs—yet the authors’ revisions satisfied most queries.

**Additional Comments On Reviewer Discussion:**

The reviewers actively participated in the discussion, and most concerns have now been addressed.

---

### Decision · Program_Chairs · 2025-01-22

Accept (Spotlight)